# The remanufacturing evaluation for feasibility and comprehensive benefit of retired grinding machine

**Tianbai Ling**⊙*, **Yongyi He**⊙

School of Mechatronic Engineering and Automation, Shanghai University, Shanghai, P.R.China

⊙ These authors contributed equally to this work.
* lingskywhite@shu.edu.cn

## Abstract

Grinding is the last and most important process of parts processing, the purpose is to achieve high precision and surface roughness. Therefore, grinding machine has the characteristics of high added value, high technology content and great remanufacturing value. However, the evaluation of machine tool remanufacturing is based on imprecise and fuzzy information at present. The aim of this study is to present the remanufacturing evaluation for feasibility and comprehensive benefit of retired grinder. Firstly, according to the unique structure of grinder, the feasibility evaluation model of grinder remanufacturing is established, including technical feasibility criterion, economic feasibility criterion and resource environment feasibility criterion. Secondly, the comprehensive benefit evaluation model of remanufacturing grinder is established, in which the weight of each evaluation criterion is determined by Analytic Hierarchy Process (AHP). Finally, combined with the remanufacturing case of the cylindrical grinder, the evaluation method is verified and analyzed. The results show that the remanufacturing of the waste grinding machine through the feasibility evaluation can obtain better comprehensive benefits, and the remanufacturer can get considerable benefits and reduce the potential risks in the remanufacturing process.

## Introduction

### Development of remanufacturing abroad

From 1930s to 1940s, American automobile manufacturing industry took the lead in remanufacturing practice. After nearly 80 years of development, the developed countries in Europe and America, represented by the United States, Germany, the United Kingdom, the Netherlands and Japan, have established a remanufacturing industry system covering many units, such as waste product recycling and disassembly detection, remanufacturing production control, remanufacturing technology research development and remanufacturing product marketing [1]. The remanufactured products cover many fields, such as aerospace, heavy off-road equipment, automotive parts, locomotives, information technology products, electrical equipment, medical equipment and office appliances. It have formed perfect remanufacturing

**Data Availability Statement:** All relevant data are within the paper.

**Funding:** Ling Tianbai (LTB) Z16030384 Shanghai Science and Technology Development Foundation (CN) http://www.stefg.org/mobile/about/inventory.

aspx?key=&hangye=&fenhui=
72634837642510336 no sponsors or funders play any role in the study design, data collection and analysis, decision to publish, or preparation of the manuscript.

**Competing interests:** The authors have declared that no competing interests exist.

operation mode and mature remanufacturing market environment. The annual output value of the U.S. remanufacturing industry rose rapidly from $75 billion in 2005 to $100 billion in 2016, The United States has become the largest country in the world in remanufacturing industry, and the annual output value of the global remanufacturing industry rose from $100 billion in 2005 to $140 billion in 2016. Therefore, the remanufacturing industry is called "giant in industry" in foreign countries [2,3].

Machine tool is a kind of power mechanical device with complex structure, which manufactures mechanical parts by metal cutting. Grinding machine, as a typical mechanical and electrical equipment, has a high value of recycling and remanufacturing. The waste grinder can be recycled and utilized as well as the iron and steel resources, thus greatly saving iron and steel resources. The cost of remanufacturing grinder can be reduced by more than 40% compared with the purchase of new machine tools, the durability of the mechanical part of grinder is strong and stable, especially for the bed and column castings, the longer the aging, the better the performance and reliability, and the grinder is suitable for recycling and remanufacturing. The product and functional components have good interchangeability, which greatly simplifies the remanufacturing process.

Remanufacturing, as a specific type of recycling, makes the fact that the used durable goods can be repaired to a condition like new realized. By means of remanufacturing, most of the used machinery parts can be repaired to a condition like new with warranty to match, which not only alleviates environmental contamination, but reduces energy consumption and professional labor used in production.the most importance is that the product performance after remanufacturing should reach or exceed the original product [4–6].

Throughout the development of remanufacturing industry abroad, the United States, Europe, Japan and other developed countries and regions of remanufacturing industry have done a lot of work in the aspects of remanufacturing technology management and practice of remanufacturing enterprises. The machine tool remanufacturing in the United States has experienced the development stages of maintenance, renovation, Numerical Control (NC) transformation and remanufacturing. After years of development and with the elimination of a large number of enterprises in the market, there are more than 300 enterprises specializing in machine tool remanufacturing in the United States, mainly third-party machine tool remanufacturing service providers [7,8]. Maintenance Service Corp. has a 60-year history of machine tool modification and remanufacturing. They can transform, renovate and remanufacture all kinds of machine tools. The company has completed more than 20,000 sets of machines in total. Machine Tool Builders, Inc. has been engaged in machine tool remanufacturing business for nearly 20 years. They mainly remanufactures gear processing machine tools. At present, The company has the ability to design and manufacture new machine tools, and has completed a number of machine tool remanufacturing business. In Japan, the manufacturing industry breaks away from the traditional maintenance concept of old machine tools and equipment in order to modernize them by remanufacturing. According to statistics, there are at least 20 enterprises engaged in machine tool remanufacturing and have a certain scale in Japan, such as Osaka Engineering Company, Nozaki Engineering Company, Yamazaki Mazak Company, etc.

## Development of remanufacturing in China

Machine tool remanufacturing industry has developed rapidly in recent years in China. The promotion by centralized environmental legislation, environmental awareness of customers has been continuously enhanced. the economic and social benefits of remanufacturing enterprises and customers have been continuously improved. However, due to the lack of key

remanufacturing technology, maintenance and modification are still the main services of machine tool remanufacturing, which can not guarantee its performance and reliability.

Since 1999, government has vigorously publicized and deeply studied remanufacturing project in China. In 2005, Jinan Fuqiang Power Co., Ltd. was recognized as a pilot unit of national circular economy, which indicates China's willingness to make strategic attempts in remanufacturing; In 2010, eleven ministries and commissions jointly issued that Opinions on Promoting the Development of Remanufacturing Industry, which is an important milestone of remanufacturing in China; In order to implement the Circular Economy Promotion Law in 2013, Chinese government strongly supports the promotion and use of remanufactured products, promotes the recycling of used remanufactured parts and enlarges the market share of remanufactured products; In 2015, the government has issued the Outline of Made-in-China 2025 and proposed to implement high-end remanufacturing and intelligent remanufacturing to promote the sustainable development of remanufacturing industry [9].

Machine tool remanufacturing enterprises mainly include original machine tool manufacturing enterprises and independent third-party machine tool remanufacturing enterprises in China. Chongqing Machine Tool Group Co., Ltd. regards machine tool remanufacturing as one of the major strategies of the enterprise, and gradually make the remanufacturing of ordinary mechanical gear hobbing machine a new growth point of enterprise profits. In addition, Shenyang Machine Tool Group, Dalian Machine Tool Group, Shanghai Machine Tool Works Co., Ltd and other companies are engaged in machine tool remanufacturing services.

Along with the rapid increase in living standard, the consumption of energy and non-renewable material is rapidly reaching, what many experts believe, unsustainable levels, which poses significant environmental challenges [10]. Remanufacturing is a powerful product recovery option which generates products as good as new ones from old discarded ones. This technique can also help to reduce the environmental impact of the product in its final disposal. Growing concern for resource conservation and waste reduction led to the augmentation of remanufacturing. This paper presents a feasibility evaluation method for grinder remanufacturing, which evaluates the technical feasibility, economic feasibility and resource environment feasibility, provides information support for grinder remanufacturing [11–14].

## Literature review

In the context of resource shortage and environmental pollution, remanufacturing as one of the well-known recovery methods for end-of-life products has become a research hotspot of various research institutions. The ability to design products for remanufacturing is usually owned by the original equipment manufacturers (OEM), who have control over both the design and remanufacturing phases of machine tool products. Not all machine tools are suitable for remanufacturing, only products with certain intrinsic value, long technology life cycle or high durability can be considered for remanufacturing. Detailed guidance for identifying remanufacturing candidates or components can be found in remanufacturing literature [15,16].

Modular design is built on the basis of functional analysis of a product with different functions or with the same function, but different performances and specifications. The use of modular design for remanufacturing is an effective way to perform the configuration of the product structure at the conceptual design stage. The modular division of machine tool can meet the requirements of remanufacturing of machine tool according to the characteristics of each stage of product life cycle, the economic and technical criteria of remanufacturing machine tools are mainly considered. In the process of modular design of machine tool

products, module division is an important aspect of modular design, and it is of great significance for enterprises to implement machine tool remanufacturing [17,18].

A multi-objective optimization model with life cycle equilibrium and cost to maximize the recovery value of end-of-life (EOL) products was established. Case analysis showed that this scheme could effectively optimize the recovery value of EOL products and improve the economic benefits of remanufacturing [19]. Different from the part remanufacturing such as automotive parts, machine tool remanufacturing is the typical machine-based business, in which the used machine tools are reused as the cores of remanufacturing to meet market demands by the processes of product innovation redesign and machine upgrading. Not only the performance of the used machine tool can be restored to like-new condition, but also the energy efficiency, ecology efficiency and information function can be improved.

The important feature of remanufacturing is that the quality of remanufactured product is not inferior to the new product, and product performance has been significantly upgraded or improved. The cost of remanufactured products is only about 50% of new products. Compared with new products, remanufactured products can save about 60% of energy and 70% of materials. The environmental impact of remanufactured products is significantly reduced compared to the manufacture of new products. Restoring retired products to their original state is just repair or overhaul, not remanufacturing [20].

From literature review and analysis, it is concluded that most of the relevant literature focuses on the modular design and redesign in new product development as well as the alternative selection of remanufacturing process and mainly include economic, technical, environmental and social factors. As a mechanical product with high value and complex structure, there is no suitable remanufacturing decision-making method, remanufacturing evaluation system and remanufacturing case analysis for selection in grinder remanufacturing. This paper establishes the feasibility evaluation and comprehensive benefit evaluation model of grinding machine remanufacturing from the combination of qualitative and quantitative methods to fill the gaps in the literature of current grinding machine remanufacturing decisions.

## Analysis of grinding machine remanufacturing process

### Introduction of grinding machine structure

The main features of high-speed Computer Numerical Control (CNC) grinder include high static stiffness and automation of the whole grinder, high processing accuracy, reliability and stability, and convenient use and maintenance. Workbench is driven by ball screw pair, the minimum setting unit is 0.1um. The grinding wheel spindle system adopts the hydrodynamic spindle system, which has the characteristics of high stiffness, high rotation accuracy, high grinding efficiency, less vibration and strong bearing capacity under working conditions. CNC grinder adopts fully enclosed external protective device, which has a high safety factor. At the same time, the internal electrical machinery chain is stable and reliable to ensure safe production.

CNC grinding machine is T-shaped overall layout, in which as an independent module, there are electric box, cooling and filtering system, hydraulic station and CNC panel control box. The overall structure of grinder consists of bed, motor, worktable, headframe, tailstock and grinding wheel spindle system. The overall three-dimensional model is shown in Fig 1.

Workbench is installed on the guide rail of the front bed of the grinder. Headframe and tailstock are installed at both ends of the workbench. According to the different processing requirements and the shape of the parts, Mohs taper top, chuck or pneumatic chuck can be installed between the headframe and tailstock. Due to the different processing lengths of

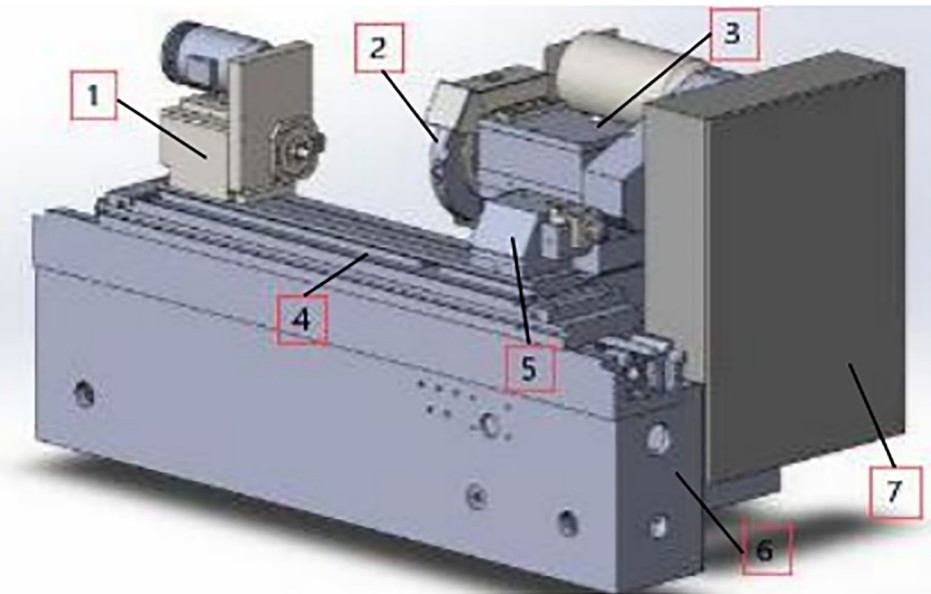

**Fig 1. The overall three-dimensional model of grinding machine.** 1: headframe; 2: grinding wheel; 3: wheelhead frame spindle system; 4: workbench; 5: tailstock; 6: bed; 7: electrical control box.

different parts, the tailstock adopts linear guide rail, which has high guiding accuracy and can be moved longitudinally for clamping operation.

The grinding wheel spindle system is installed on the guide rail of the back bed of the grinder, and the spindle system is the most critical core component of the grinder. The spindle system can move on the transverse guide rail to realize the axial feeding motion and to adjust the processing distance between the grinding wheel and the workpiece conveniently. The grinder spindle is supported by hydrodynamic and hydrostatic bearings, the spindle system is equipped with an independent lubrication device, which improves the life of bearings and spindles.

Because the grinder structure needs to meet the requirements of high processing accuracy, high rigidity and strong bearing capacity, the grinder guideways are basically in the form of integrated guideways that is hard rails. As shown in Fig 2, the guide rails and the bed are integrated casting parts, which are processed on the basis of the bed castings, and then processed into guideways by quenching and grinding. Machine tools such as lathes, milling machines and machining centers are basically in the form of linear guideways. As shown in Fig 3, linear guides and machine bed are separated structures. After the guides are worn out, they can be disassembled and replaced. The structure is simple and the processing is convenient, but the rigidity and bearing capacity of linear guides are worse than that of hard ones.

## Grinding machine remanufacturing process

The model of grinder remanufacturing can be divided into contract remanufacturing and commercial remanufacturing. Because there are many uncertainties in the quality grading process of recycled waste grinders, the remanufacturing process is more complex than that of new machine tools. Therefore, in the remanufacturing market of China, the remanufacturing grinders is mainly based on contract remanufacturing mode, which provides corresponding remanufacturing machine tools according to customer needs [21].

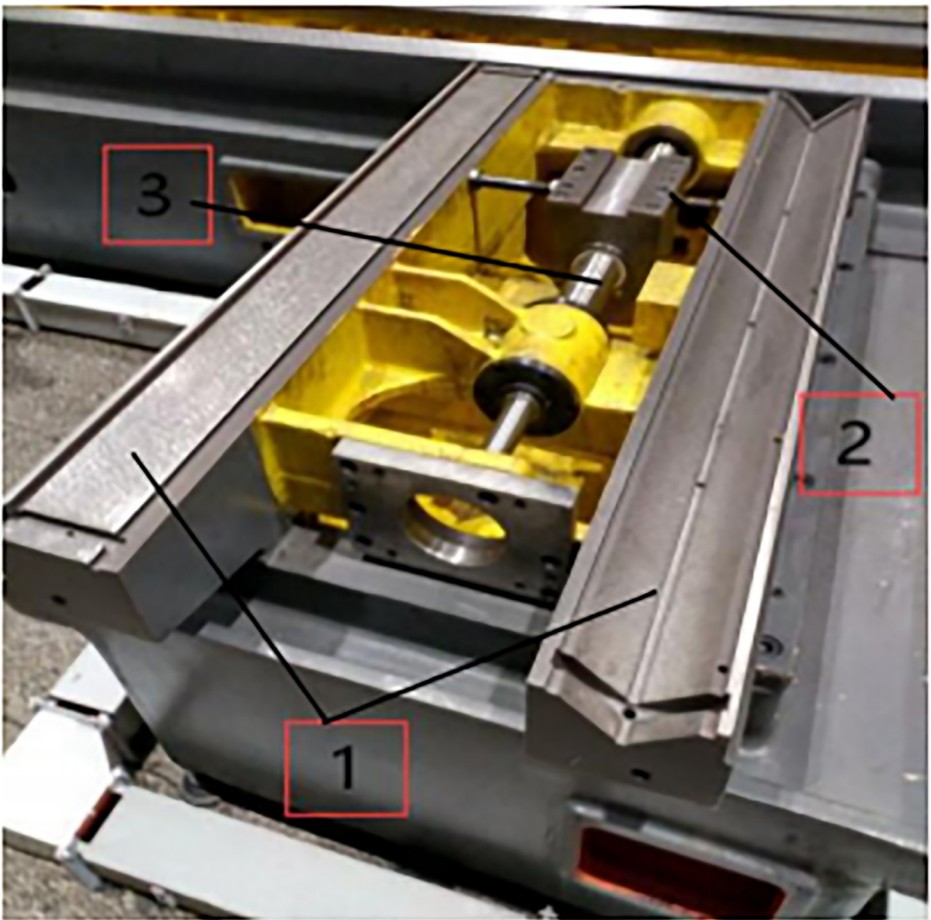

**Fig 2. Integrated guideways.** 1: integrated bed guideways; 2: nut seat; 3: ball screw.

Due to the limitation of existing structure and material, the remanufacturing process of waste grinder must be adapted to the current situation of products. As shown in Fig 4, grinder remanufacturing is a systematic organization. Firstly according to the flow chart of remanufacturing production of waste grinder, it is determined that the machine tool can meet the remanufacturing standard before it can be recycled. Secondly the process steps of disassembly, cleaning, inspection and classification are implemented for the parts of waste machine tool that meet the remanufacturing evaluation standard, and the remanufactured parts of waste machine tool are stored in the storage area to be processed. Finally the surface treatment technology is used to re-process the remanufactured parts. The remanufactured parts and the purchased new parts are reassembled and the remanufactured grinder is generated. The scrap grinder parts after disassembly, cleaning and testing can be divided into four categories according to their damage status: direct reusable parts, remanufactured parts, recyclable parts and direct scrap parts [22,23].

Through the remanufacturing of grinders, the residual value of waste grinders can be fully reused. The performance of remanufacturing grinders can be equal to or better than that of new machine tools, and the functions that new machine tools do not have can be increased, but the cost is only 40% to 70% of new machine tools.

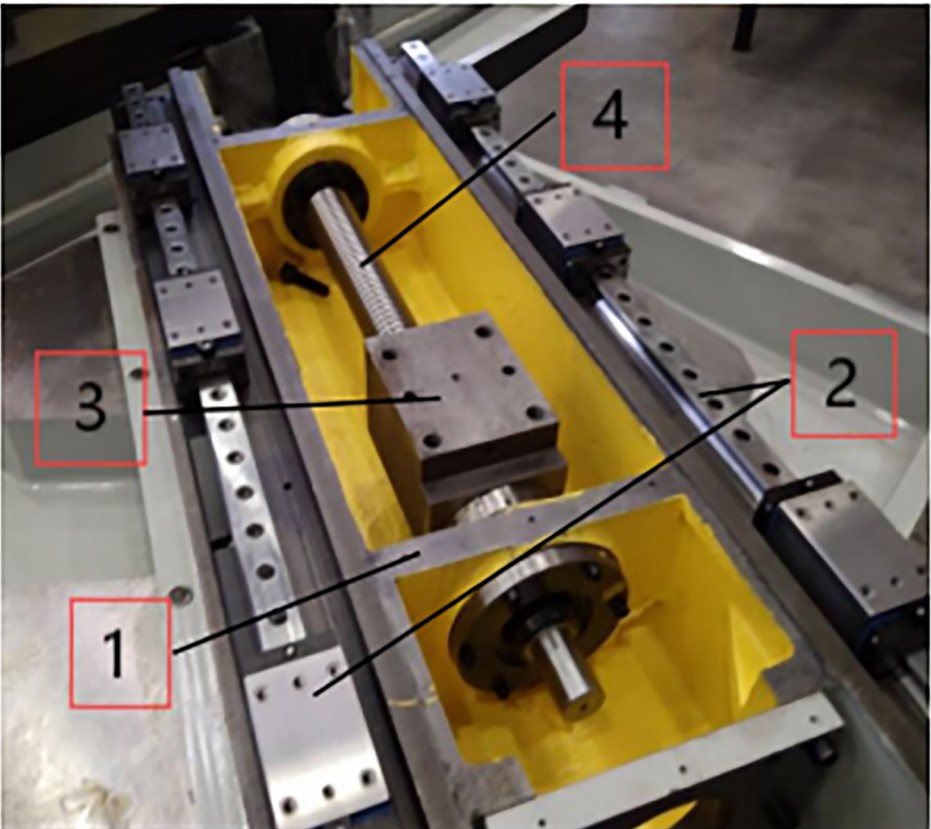

**Fig 3. Linear guideways.** 1: bed; 2: linear guideways; 3: nut seat; 4: ball screw.

## Feasibility evaluation for the remanufacturability of retired grinding machine

### Evaluating process of grinding machine remanufacturability

Based on the analysis of grinder remanufacturing process model, this paper establishes a comprehensive evaluation method model of the remanufacturing feasibility of waste grinders based on expert experience knowledge, as shown in Fig 5. Firstly, considering the serious damage and wear of the recycled waste grinder, the remanufacturing of the grinder may be difficult to implement, so it is necessary to analyze the technical feasibility of the remanufacturing of the waste grinder to ensure that the machine can use modern manufacturing, information, numerical control and automation technology to achieve its functional recovery and performance improvement. If the technical feasibility of remanufacturing of the grinder is poor, then only the material recovery and recycling treatment can be carried out. Secondly, the cost of remanufacturing of the waste grinder with better technical feasibility needs to be calculated and evaluated. It is necessary to determine whether the remanufacturing of the grinder can make the remanufacturer profitable. If not, the remanufacturer will adopt material recovery and other methods to deal with the remanufacturing cost. The third step is to analyze the resource and environmental benefits of grinder remanufacturing from the perspectives of material saving, energy saving and environmental emission reduction. The purpose of remanufacturing is to save resources and reduce emissions by recycling and reusing waste resources.

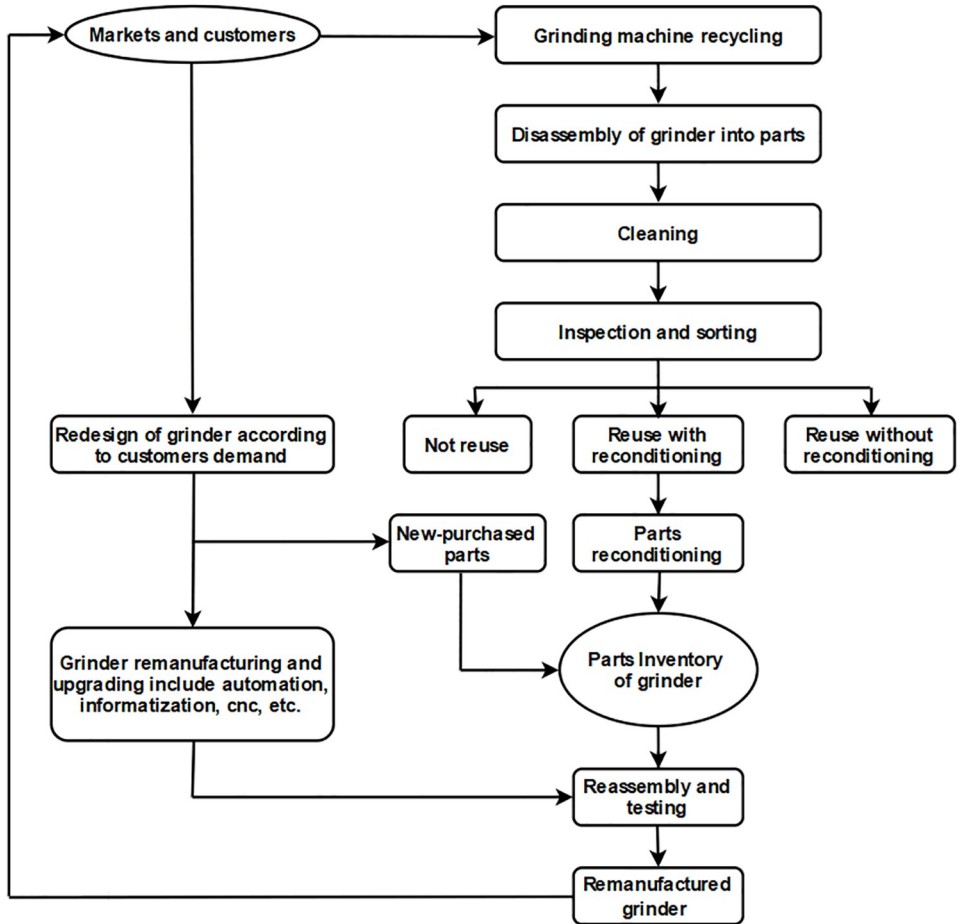

**Fig 4. Process system of grinding machine remanufacturing.**

Finally, the waste grinder with better remanufacturing feasibility enters disassembly, cleaning, testing and classification. In the process of repairing, reprocessing and reassembling, the performance level of new machine tools can be achieved by means of precision recovery, function recovery and performance improvement, while the machine tools with poor feasibility of remanufacturing can be recycled by means of material recovery.

On the basis of data investigation and case analysis of grinder remanufacturing industry, it is concluded that the feasibility evaluation of waste grinder remanufacturing is mainly considered from three aspects which is technical feasibility, economic feasibility, resource and environment feasibility. The feasibility evaluation system of grinder remanufacturing is shown in Table 1. The evaluation criteria are mainly composed of three parts, that is technical feasibility, economic feasibility and resource and environment feasibility. Each criterion can be subdivided into various indicators. The evaluation value including technical feasibility, economic feasibility, resource and environment feasibility is between 0 and 1. When the evaluation value is between 0.6 and 0.74, it means that the machine tool can be remanufactured. When the evaluation value is 0.75 to 0.89, it indicates good, and the remanufacturing of machine tool is profitable. When the evaluation value is more than 0.9, it means excellent, and machine tool remanufacturing can produce great benefits. The definition and calculation of each evaluation value are as follows.

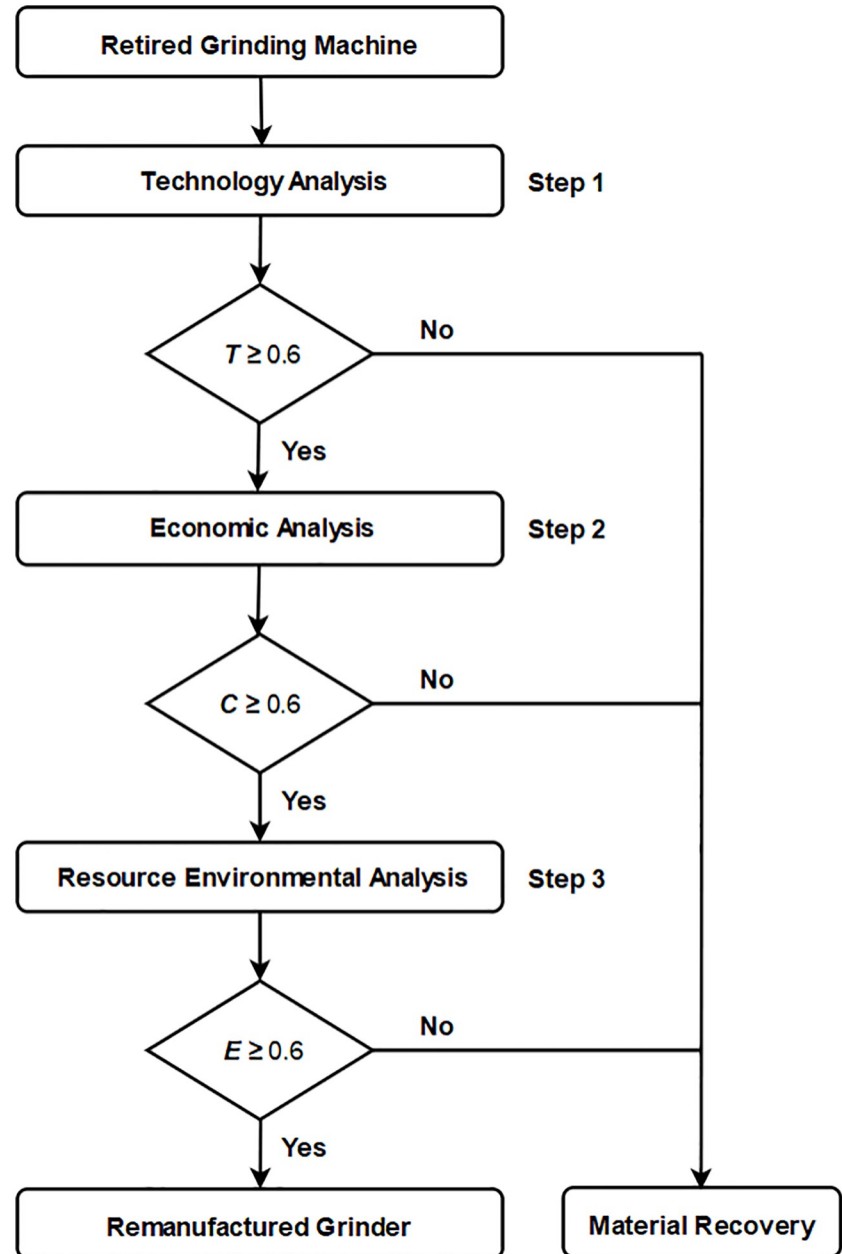

**Fig 5. Evaluating process of grinding machine remanufacturability.**

## Criteria of technology feasibility

The technological feasibility of remanufacturing of waste grinders should be considered from all stages of remanufacturing process, including the condition of the bed guide of casting, the simplicity of disassembly, the feasibility of cleaning, the feasibility of detection and classification, and the feasibility of parts and components remanufacturing. Disassembly is the first step in the remanufacturing process and the basis and premise for the scrap grinder to enter the remanufacturing process. Because remanufacturing is not considered in the original design

**Table 1. Feasibility evaluation system of grinder remanufacturing.**

| Criteria | Index | Notions |
|---|---|---|
| Technology feasibility | Ease of disassembly | $t_d$ |
| | Feasibility of cleaning | $t_c$ |
| | Feasibility of inspection and sorting | $t_i$ |
| | Feasibility of part reconditioning | $t_r$ |
| Economic feasibility | Comprehensive economic feasibility | $C$ |
| Resources environment feasibility | Material and energy saving | $E$ |

process of the waste grinder, parts of the waste grinder may be damaged in the disassembly process.

The disassembly of grinder remanufacturing is different from material recovery and recycling [24]. The damage of parts caused by disassembly should be minimized and the disassembly efficiency should be improved. There are various kinds of bolts, pins and other connections on the machine tool, these connections have different disassembly simplicity. The simplicity of disassembly is a qualitative index, which is generally evaluated by experts before remanufacturing from the aspects of connection structure, connection mode, number of parts and components. In order to simplify and quantify the disassembly simplicity, this paper mainly evaluates the disassembly time, as shown in Formula 1 and 2.

$$\delta = \frac{\sum_{i=1}^{N} c_i \times t_i}{T_d} \quad (i = 1, 2, \cdots, N) \tag{1}$$

$$t_d = \begin{cases} 1.00 & \delta \leq 1.0 \\ 0.85 & 1.0 < \delta \leq 1.2 \\ 0.60 & 1.2 < \delta \leq 1.4 \\ 0 & 1.4 < \delta \end{cases} \tag{2}$$

Among them, $t_i$ represents the average disassembly time of the $i$th connecting parts, $c_i$ represents the number of the $i$th connecting parts, $N$ represents the number of connecting parts, $T_d$ represents a standard value of disassembly time, $\delta$ is an intermediate variable, and $t_d$ represents the disassembly simplicity index value.

In order to make the recycled decommissioned grinders and parts have better reprocessability, the disassembled parts must be cleaned before detection and classification. Different parts need different cleaning methods, and different cleaning methods have different degrees of difficulty. In this paper, the feasibility of cleaning is mainly analyzed from the number of parts and the difficulty of cleaning methods. Based on the investigation of remanufacturing enterprises, it is assumed that the degree of difficulty of various cleaning methods, such as blowing or brushing, baking, chemical detergent spraying and ultrasonic cleaning is 0.20, 0.40, 0.60 and 0.80. The cleaning feasibility index can be calculated by formula 3.

$$t_c = 1 - \frac{\sum_{j=1}^{4} L_j \times \theta_j}{\sum_{j=1}^{4} L_j} \quad (j = 1, 2, 3, 4) \tag{3}$$

Among them, $L_j$ represents the number of parts that are cleaned by the $j$th washing method, and $\theta_j$ represents the difficulty value of the $j$th washing method.

Performance testing of waste grinder parts is the key link of remanufacturing grinder quality, in order to ensure the quality of remanufacturing grinder, it is necessary to inspect all dismantled parts to ensure that they are not damaged and reusable. Based on the test results of parts, parts can be divided into three categories, that is direct use without reprocessing, usable and unusable after reprocessing. In order to simplify the calculation difficulty, the index value is mainly evaluated by the average detection time and judged by experts. The evaluation result $t_i$ can be divided into four grades (A, B, C, D), and the corresponding index values are (0.95, 0.82, 0.65, 0.50).

Repairing and reprocessing is the key link to restore the waste grinder parts to the new product, it is the most important process of grinder remanufacturing process [25]. Because different users have different working conditions and different usage habits, the quality level of recycled parts varies greatly. Waste grinder parts may have problems in the process of reprocessing and can not continue to use, only by new purchase or new parts to replace. Different parts have different reprocessing processes. Each process has different costs and different success rates. The cost of grinder remanufacturing directly depends on the reuse rate of waste parts. If the waste grinder has good feasibility of parts remanufacturing, it will significantly improve the economy of machine tool remanufacturing. The feasibility index of repairing and reprocessing waste parts can be calculated by formula 4.

$$t_r = \frac{\sum_{k=1}^{N} Q_k \times p_k}{\sum_{k=1}^{N} Q_k} \quad (k = 1, 2, \cdots, N) \tag{4}$$

Among them, $Q_k$ represents the number of $k$th parts after disassembly and cleaning, $p_k$ represents the success rate of repair and reprocessing of $k$th parts, and $N$ represents the number of parts.

Because the grinder bed and guide rail adopt an integrated structure, the grinder can not be manufactured when the guide rail is seriously worn or deformed and cracks occur, so it can directly recycle resources. Based on the above analysis of each index of technical feasibility and grinder structure form, the evaluation value of technical feasibility can be obtained according to formula 5.

$$T = \alpha(\eta_d t_d + \eta_c t_c + \eta_i t_i + \eta_r t_r) \tag{5}$$

Among them, $\eta_d$, $\eta_c$, $\eta_i$ and $\eta_r$ represent the weight values of $t_d$, $t_c$, $t_i$ and $t_r$ respectively, the weight values can be determined by AHP. The value of $\alpha$ is 0 or 1. When $\alpha = 1$, it means the bed guide rail can be recycled after repair; When $\alpha = 0$, it means the bed guide rail is severely worn or corroded and can not be recycled.

## Criteria of economic feasibility

From the aspect of economic feasibility, whether the waste grinder is suitable for remanufacturing is mainly analyzed in terms of making full use of its added value according to the existing technology and technology level to obtain economic benefits. The greater the added value and the lower the remanufacturing cost, the better the economic feasibility and the higher the degree of remanufacturing. In addition, grinder remanufacturing also has significant indirect economic benefits, which can save a large amount of equipment investment and technical training costs of customer enterprises, greatly reduce the cost of purchasing new machine tools of the same performance level, shorten the delivery time by half, and the performance can reach or exceed the original level of new machine tools. This paper mainly judges the

economic feasibility of remanufacturing from the cost of grinder remanufacturing, and evaluates the economic benefits of grinder remanufacturing.

This paper divides the economic feasibility evaluation index $C$ into two parts, one is to measure the cost of remanufacturing $C_1$, the other is to measure the cost of new machine tool manufacturing $C_2$, the specific measurement indicators are as follows.

Grinding machine remanufacturing cost $C_1$ mainly includes machine tool recovery, disassembly, cleaning, testing, reprocessing and reassembly costs. Each cost mainly includes management cost, material cost, resource cost and human cost. The new manufacturing cost $C_2$ of grinding machine mainly includes raw material cost, processing cost, assembly cost and quality inspection cost. In the evaluation of the new manufacturing sector, the cost of raw materials includes not only the cost of processing raw materials, but also the purchasing cost of the parts purchased from machine tools.

The evaluation method of economic indicators is different from that of technical indicators. It belongs to quantitative evaluation. Therefore, economic indicators can not be evaluated by analytic hierarchy process. According to Lund Robert in document, if the cost of remanufactured products is 40% to 70% of the price of new products with the same performance, the remanufacturer will benefit. The cost ratio of remanufacturing to new manufacturing is an important parameter to measure the economic feasibility.

Firstly, objective data are obtained by field investigation and evidence collection according to the established indicators, and then the cost $C_1$ and $C_2$ under each index of remanufacturing and new manufacturing are obtained by statistical collation of the data.

Secondly, the ratio $f$ of $C_1$ to $C_2$ is calculated.

$$f = \frac{C_1}{C_2} \tag{6}$$

Finally, according to the different values of $f$, the economic feasibility evaluation index $C$ is obtained.

$$C = \begin{cases} 1 & f < 0.4 \\ 1.8 - 2f & 0.4 \leq f \leq 0.7 \\ 0 & 0.7 < f \end{cases} \tag{7}$$

When $f < 0.4$, that means the remanufacturing cost is less than 40% of the new manufacturing cost, and the profit of remanufacturing is larger, so the economic feasibility index is 1, which means it is very feasible. When $0.4 < f < 0.7$, the remanufacturing cost is 40% to 70% of the new manufacturing cost of similar grinder products, remanufacturing will be profitable, but the profit margin is inversely proportional to the proportional coefficient. When $0.7 < f$, the remanufacturing cost is more than 70% of the manufacturing cost of the same new machine tool. At this time, the profit margin of remanufacturing is almost zero, so the economic feasibility index is 0, indicating that it is not feasible.

## Criteria of resource environment feasibility

The feasibility of grinder remanufacturing is mainly embodied in three aspects which is resource value, energy value and impact on the environment of the waste grinder. The resource utilization rate, energy saving rate and environmental protection rate are used to evaluate the feasibility of grinder remanufacturing.

The resource utilization rate of remanufacturing resources refers to the parts that are assembled directly by grinding machine. The raw material of remanufacturing grinder comes

from waste parts, so its resource utilization rate is reflected in the use of remanufacturing of waste parts. The evaluation index of resource utilization rate of remanufacturing grinder is shown in formula 8.

$$R = \frac{N_u}{N_t} \times \frac{V_u}{V_t} \tag{8}$$

Among them, $N_t$ is the total number of parts for the new grinder, $N_u$ is the number of waste parts for the remanufacturing grinder, $V_t$ is the total value of the new grinder, $V_u$ is the value of the remanufacturing grinder, $N_u/N_t$ is the proportion of the waste parts, $V_u/V_t$ is the utilization value of the waste parts.

The energy saving rate of grinder remanufacturing refers to the amount of energy saved in the process of remanufacturing compared with the energy needed in the new grinder manufacturing. Energy refers to the energy consumed by water, electricity and coal for parts repair and replacement in remanufacturing process. It indirectly participates in the composition of remanufactured products. The evaluation index of remanufacturing energy saving rate is shown in formula 9.

$$P = \frac{1}{n} \sum_{j=1}^{n} \frac{V_1 - V_2}{V_1} \tag{9}$$

Among them, $n$ is the number of types of energy needed, $V_1$ is the amount of energy needed for new manufacturing grinders, and $V_2$ is the amount of energy needed for remanufacturing grinders.

Different from other indicators, it is not easy to quantitatively describe the impact of remanufacturing feasibility of waste grinders on the environment. The purpose of establishing this indicator is to analyze the environmental feasibility of remanufacturing grinders. The evaluation method is to compare the pollutants discharged from remanufacturing process with the national environmental protection standards, and to establish the evaluation index function of environmental protection rate as shown in formula 10.

$$\eta = \begin{cases} 1 & x \leq q \\ 0 & x > q \end{cases} \tag{10}$$

Among them, $x$ is the discharge of various pollutants in remanufacturing process and $q$ is the discharge of pollutants that meet the national emission standards.

According to the index of resource utilization rate, energy saving rate and environmental protection rate, the comprehensive index $E$ of resource environment evaluation index can be obtained, as shown in formula 11.

$$E = \eta(\omega_1 R + \omega_2 P) \tag{11}$$

Among them, $\omega_1$ and $\omega_2$ mean the weight of resource utilization and energy saving rate. According to experience, they are 0.75 and 0.25 respectively.

## Determining weights for individual index by AHP

In order to determine the evaluation value of individual index of grinder remanufacturing, the weight value of individual index must be determined first. It is impossible to obtain absolute weights, but only relative weights. AHP is a decision-making method that decomposes the relevant elements into objectives, criteria and schemes, on which qualitative and quantitative analysis can be carried out [26–28].

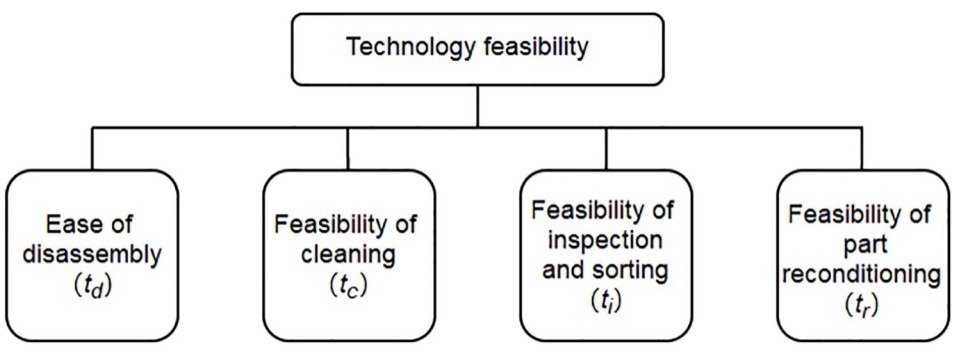

**Fig 6. AHP model of technology feasibility.**

Because different customers have different requirements for the remanufacturing of used grinders, the weights of individual index are subjective and may vary according to different customer weights. According to the characteristics of AHP method, it is applied to determine the weight of index layer in the feasibility of grinder remanufacturing. The expert platform is composed of customers and the design and management personnel of remanufacturing machine tools. The process of determining the weight of individual index by AHP method is as follows.

Firstly, aiming at the problem of determining the weight of technology feasibility index system, an analytic hierarchy process model is established, as shown in Fig 6.

Secondly, it establishes the priority judgment scale of technical feasibility indicators, and the value can be obtained by 1 to 9 scale method, as shown in Table 2.

According to the judgment scale, individual technical feasibility index is compared with each other, and the judgment matrix is established, as shown in Table 3.

Based on AHP, the relative weight of the feasible index of grinder remanufacturing technology can be calculated by formula 12.

$$\omega_i = \frac{\overline{W_i}}{\sum\limits_{i=1}^{n} \overline{W_i}} \qquad (i = 1, 2, \cdots, n) \qquad (12)$$

Among them, $\overline{W_i} = \sqrt[n]{M_i}$ , $M_i = \prod\limits_{j=1}^{n} t_{ij}$.

**Table 2. Priority judgment scale of technical feasibility elements.**

| Intensity of importance | Definition | notions |
|---|---|---|
| 1 | Equal importance | $t_i$ and $t_j$ contribute equally to the objective |
| 3 | Moderate importance | $t_i$ is slightly favored over $t_j$ |
| 5 | Obvious importance | Experience and judgment strongly favor $t_i$ over $t_j$ |
| 7 | Very strong importance | $t_i$ is favored very strongly over $t_j$ |
| 9 | Extreme importance | The evidence favoring $t_i$ over $t_j$ is of the highest possible order of affirmation |

Intensities of 2, 4, 6, and 8 can be used to express intermediate values

**Table 3. Judgment matrix of technical feasibility index.**

| | Ease of disassembly ($t_d$) | Feasibility of cleaning ($t_c$) | Feasibility of inspection and sorting ($t_i$) | Feasibility of part reconditioning ($t_r$) |
|---|---|---|---|---|
| Ease of disassembly ($t_d$) | 1 | 3 | 5 | 1 |
| Feasibility of cleaning ($t_c$) | 1/3 | 1 | 2 | 1/3 |
| Feasibility of inspection and sorting ($t_i$) | 1/5 | 1/2 | 1 | 1/5 |
| Feasibility of part reconditioning ($t_r$) | 1 | 3 | 5 | 1 |

Finally, the weights of individual technical feasibility index are determined by AHP, and the results are shown in formula 13.

$$W = \{t_d, t_c, t_i, t_r\} = \{0.3937, 0.1374, 0.0752, 0.3937\} \tag{13}$$

## Comprehensive benefit evaluation for grinder remanufacturing

### Contents of comprehensive benefit evaluation

The comprehensive benefit evaluation of grinder remanufacturing is a complex system with fuzziness. The benefit of grinder remanufacturing is hierarchical. The technical benefit of remanufacturing can be divided into product quality, timeliness of remanufacturing and upgrading of product technology. Because the comprehensive benefit evaluation involves many factors, but lacks data, many factors are in the state of fuzzy qualitative analysis, so the comprehensive benefit evaluation of grinder remanufacturing can adopt the fuzzy AHP [29].

The evaluation of grinder remanufacturing benefits should be based on technical benefits, including technical benefits, economic benefits, environmental and resource benefits. Technical benefit refers to product energy efficiency, remanufacturing product quality and remanufacturing time after remanufacturing activities, including comprehensive technical upgrading of grinders after remanufacturing. Economic benefit refers to the benefit of remanufacturing activities in product life-cycle cost management. Environmental benefit refers to the resource saving and environmental protection benefits of remanufacturing activities.

The evaluation system of remanufacturing benefits studied in this paper is divided into two levels. The first level evaluation index is technical, economic and environmental benefits of grinder remanufacturing, which constitutes the first level evaluation factor set $U$.

$$U = (u_1, u_2, u_3) \tag{14}$$

Based on the analysis of grinder remanufacturing practice, the technical, economic and environmental benefits are subdivided into two levels to form a set of secondary evaluation factors. The comprehensive benefit evaluation system of grinder remanufacturing is shown in Table 4.

According to the experience of grinder remanufacturing and the actual needs, the commentary set $V$ is determined.

$$V = (v_1, v_2, v_3, v_4) \tag{15}$$

The comment set $V$ is divided into four grades, $v_1$ is excellent, $v_2$ is good, $v_3$ is general, and $v_4$ is poor.

The evolution process of product technology based on theory of inventive problem solving (TRIZ) is similar to the growth process of species. It has to go through four stages: infancy,

**Table 4. Comprehensive benefit evaluation system of grinder remanufacturing.**

| Target (U) | First level index ($u_i$) | Second level index ($u_{ij}$) |
|---|---|---|
| Comprehensive benefit of remanufacturing grinder | Technical benefit | Modularization level |
| | | Standardization level |
| | | Ease of reassembly |
| | | Machine upgrading |
| | | Technical maturity |
| | Economic benefit | Remanufacturing cost |
| | | Remanufacturing profit |
| | | Maintenance cost |
| | Environmental benefit | Material reuse rate |
| | | Energy conservation rate |

growth, maturity and withdrawal. These four stages constitute the technology life cycle of products. TRIZ theory studies the product as a technology system. Through the evaluation of the current product technology, it predicts which stage the current product is in the technology life cycle [30].

Remanufacturing grinder technical maturity refers to the stage of remanufacturing grinder products in technology, whether they can meet customer needs and market requirements. In this paper, the stage of the technical life cycle of remanufacturing grinder is predicted by analyzing the output of machine tools and patent data. Firstly, patent data is retrieved and screened. Secondly, the number of patents and the sales volume of products are analyzed. Finally, the technology maturity of products is predicted and evaluated.

As for the prediction of the technical maturity of the remanufacturing grinder, it can be seen from CNIPA that there are 80315 patents related to machine tools in China from 2000 to 2019, as shown in Table 5. From 2000 to 2006, the number of patents has been growing steadily. From 2007 to now, the number of patents has been growing rapidly. Machine tool products are developing towards numerical control, automation and intelligence.

In 2006, the number of CNC metal cutting machine tools in China reached 85,700. Since then, the production and CNC rate of machine tools have increased year by year. The CNC rate of machine tools has increased from 15.2% in 2006 to 40.1% in 2017, the specific data are shown in Table 6 [31].

According to the statistics of machine tool patents, machine tool output and CNC rate of machine tool, the number of machine tool patents increased steadily from 2000 to 2006, it can

**Table 5. Patents on machine tools from 2000 to 2019.**

| Years | 2000 | 2001 | 2002 | 2003 | 2004 | 2005 | 2006 | 2007 | 2008 | 2009 |
|---|---|---|---|---|---|---|---|---|---|---|
| Number of patents | 196 | 167 | 203 | 264 | 301 | 429 | 524 | 850 | 1204 | 1591 |
| Years | 2010 | 2011 | 2012 | 2013 | 2014 | 2015 | 2016 | 2017 | 2018 | 2019 |
| Number of patents | 2343 | 3326 | 5124 | 6286 | 6600 | 8017 | 8125 | 8759 | 12782 | 13224 |

**Table 6. Annual output of CNC metal cutting machine tools in China from 2006 to 2017.**

| Years | 2006 | 2007 | 2008 | 2009 | 2010 | 2011 | 2012 | 2013 | 2014 | 2015 | 2016 | 2017 |
|---|---|---|---|---|---|---|---|---|---|---|---|---|
| Output of metal cutting machine tools (10000 sets) | 56.4 | 60.7 | 61.7 | 60.7 | 73.4 | 84.6 | 79.7 | 72.6 | 71.1 | 57.5 | 61.1 | 64.3 |
| CNC machine proportion (%) | 15.2 | 18.9 | 19.4 | 23.1 | 29.8 | 30.2 | 26.1 | 28.3 | 36.4 | 40.8 | 40.9 | 40.1 |

be seen that the manual grinder technology has entered the technical maturity stage in 2007 and gradually entered the withdrawal stage. Therefore, the return of the manual grinder to its original state can not obtain considerable benefits. The number of patents on intelligent and compound machine tools has been growing rapidly and the rate of CNC machine tools has been increasing since 2008, the CNC machine tool technology is still in the growth stage. Through the grinding machine remanufacturing, the CNC system is added to improve the automation of the machine tool and meet the needs of customers.

## Comprehensive benefit evaluation system

The evaluation of comprehensive benefit weight set of grinder remanufacturing can be determined by combining AHP with Expert Investigation Method (Delphi), which is similar to the determination of technical feasibility index weight in Chapter 3.5. By comparing the importance of each level of indicators in the comprehensive benefit evaluation model, the judgment matrix $A_i$ of each level of evaluation indicators is obtained, and then the maximum eigenvalue $\lambda_i$ and corresponding eigenvector $P_i$ of each judgment matrix are obtained, the consistency test of $\lambda_i$ is carried out. The $CI$ and $CR$ are shown in the formula 16, $RI$ is the average random consistency index, the values are shown in Table 7.

$$\begin{cases} CI = \dfrac{\lambda_i - n}{n-1} \\ CR = \dfrac{C}{RI} \end{cases} \tag{16}$$

When $CR$ is less than 0.1, the consistency of the judgment matrix is acceptable. Otherwise, Delphi method should be used again to compare the importance and construct the judgment matrix until $CR$ is less than 0.1. Finally, the weight set of the first-level index and the weight set of the second-level index are obtained.

For each factor $u_i$, it evaluates each aspect of comment set $V$ separately, forming a single factor evaluation fuzzy subset. Expert evaluation method can be used to determine the membership degree of each index. Several experts in the field of remanufacturing can be invited to form an evaluation expert group, and their evaluation can be indicated by scoring. $r_{ij}$ is the membership degree of any index $u_i$ in the index set $U$ to the elements in the comment set $V$, their relations are as follows.

$$r_{ij} = \frac{c_{ij}}{\sum\limits_{j=1}^{m} c_{ij}}, i = 1, 2, \cdots, n \tag{17}$$

Among them, $\sum\limits_{j=1}^{m} c_{ij}$ represents the number of expert groups, $c_{ij}$ means the number of votes in favour of factor $u_i$ and evaluation $v_j$, and the single factor membership matrix $R_i$ can be

**Table 7. $RI$ values of order 1 to 7 matrices.**

| Order number | 1 | 2 | 3 | 4 | 5 | 6 | 7 |
|---|---|---|---|---|---|---|---|
| $RI$ | 0.00 | 0.00 | 0.58 | 0.90 | 1.12 | 1.24 | 1.32 |

obtained.

$$R_i = \begin{pmatrix} r_{11} & \cdots & r_{1m} \\ \vdots & \cdots & \vdots \\ r_{n1} & \cdots & r_{nm} \end{pmatrix} \tag{18}$$

The weight vector of the first grade index is $W_i$ by single factor fuzzy evaluation, and the comprehensive benefit evaluation result $B_i$ of the first grade grinder remanufacturing can be obtained by using the weighted sum algorithm.

$$B_i = W_i \ R_i \tag{19}$$

The comprehensive benefit evaluation of fuzzy remanufacturing was calculated, the three sets of $B_i$ are taken as a subset of the first-level comprehensive evaluation, and the fuzzy comprehensive evaluation matrix $R = (B_1, B_2, B_3)^{\mathrm{T}}$ is constructed. Then the mathematical model of fuzzy comprehensive benefit evaluation for grinder remanufacturing is as follows.

$$B = W \ R \tag{20}$$

The percentile system is used to assign the four grades of the comment set $V$. Among them, $v_1$ is excellent and the assignment score is 90 to 100, $v_2$ is good and the assignment score is 80 to 89, $v_3$ is general and the assignment score is 65 to 79, $v_4$ is poor and the assignment score is 50 to 64. Quantitative value of fuzzy comprehensive benefit evaluation for remanufacturing was obtained, and the performance level of the remanufactured grinder was determined according to the remanufacturing comment set $V$.

## Results and discussion

### Feasibility analysis of grinder remanufacturing

At present, there are more than 8 million machine tools in China, but the overall level of domestic machine tools is relatively backward. According to statistics, more than 60% of machine tools have been in service for more than 10 years. These machine tools may face scrap, idle or functional elimination in the next 5 to 10 years, thus forming a considerable potential resource for machine tool remanufacturing.

In an engine parts manufacturer, Three sets of M1332B cylindrical grinders made by Shanghai Machine Tool Works Company have been in service for more than 15 years and are facing scrap and technology elimination, The retired M1332B cylindrical grinder is shown in the Fig 7. The company wants to meet the processing needs and save labor costs by purchasing CNC cylindrical grinders of the same specifications. However, due to the high price of CNC cylindrical grinders, the company can not afford to pay for three sets of new grinders. Because of the high residual value of waste cylindrical grinders, enterprises choose to remanufacture grinders to meet production needs. The cylindrical grinder is the main equipment of the enterprise. The original machine tool uses relay to control logically, the worktable is driven by hydraulic cylinder. The control technology of grinders is backward and the failure rate of the machine tool is high, but the mechanical accuracy is good. According to the requirements of enterprises, the grinder after remanufacturing is controlled by CNC system, and the feed mode of worktable is ball screw feed system. The following is the detailed process of feasibility analysis for remanufacturing of cylindrical grinder.

Firstly, the technical feasibility of remanufacturing of waste M1332B cylindrical grinder is analyzed. The first step in the remanufacturing process is to dismantle the used grinding

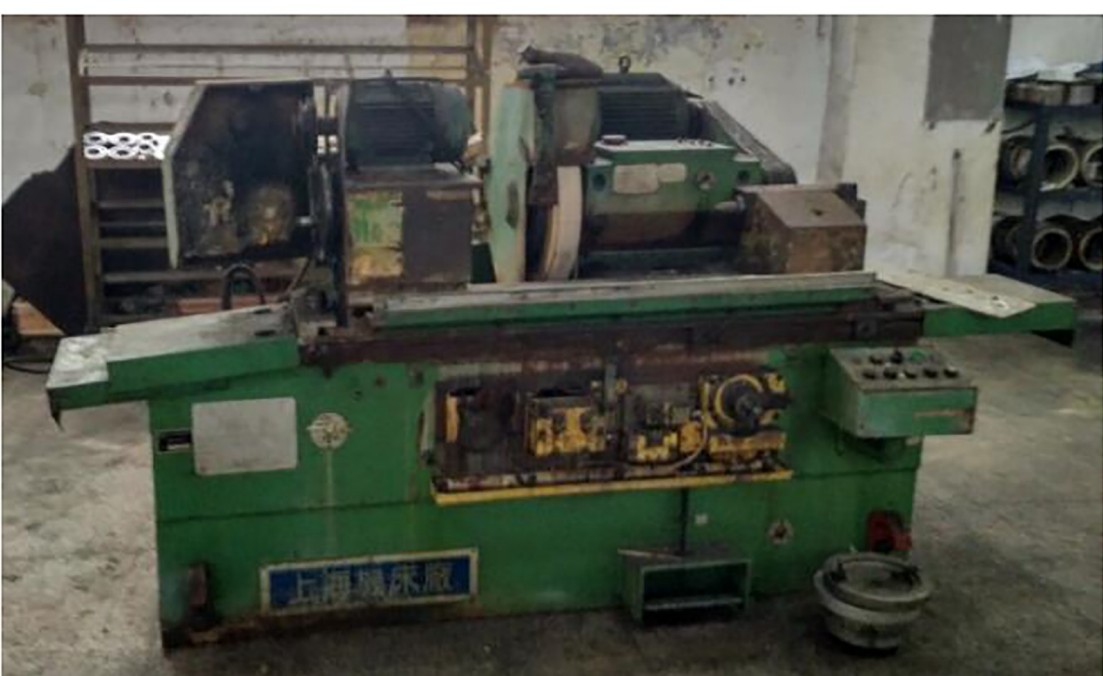

**Fig 7. The retired M1332B cylindrical grinding machine.**

machine M1332B. The main connecting parts of the machine are divided into three categories: screws, bolts and locating pins. The total number of screws is 326, with an average removal time of 5.7 seconds. The total number of bolts is 34, with an average removal time of 5.2 seconds. The total number of locating pins is 21, with an average removal time of 7.6 seconds. According to the expert's experience, the reference value of the dismantling time of the machine tool is 2,100 seconds, and the calculation results according to formula 1 are as follows.

$$\delta = \frac{c_1 \times t_1 + c_2 \times t_2 + c_3 \times t_3}{T_d} = \frac{326 \times 5.7 + 34 \times 5.2 + 21 \times 7.6}{2100} = 1.045$$

Taking the above calculation results into formula 2, $t_d = 0.85$ can be obtained.

After disassembly, the grinder has a total of 107 parts, and 7 parts such as the body shell of the grinder need to be cleaned with chemical detergent spraying, the degree of difficulty is 0.7; The total number of spindles and other shaft parts is 8, which need to be baked, and the degree of difficulty is 0.4; The other 92 parts can be treated by blowing or brushing, and the degree of difficulty is 0.2. The value of $t_c$ can be obtained by formula 3.

$$t_c = 1 - \frac{L_1 \times \theta_1 + L_2 \times \theta_2 + L_3 \times \theta_3}{L_1 + L_2 + L_3} = 1 - \frac{92 \times 0.2 + 8 \times 0.4 + 7 \times 0.7}{92 + 8 + 7} = 0.75$$

Most of the disassembled machine parts can be reused with only a small amount of wear, and the detection time is relatively small, so the evaluation result is defined as level B, which is $t_i = 0.82$.

A total of 107 machine parts need to be reprocessed after disassembly and cleaning. The total number of 31 casting parts is only a small amount of wear, and the repair success rate is 0.95; The number of 24 shaft and disk parts need to be precision reprocessed, with a success

rate of 0.85; The other 52 parts include easily deformed parts such as protective cover, with a success rate of 0.75. The value of $t_r$ can be obtained by formula 4.

$$t_r = \frac{Q_1 \times p_1 + Q_2 \times p_2 + Q_3 \times p_3}{Q_1 + Q_2 + Q_3} = \frac{31 \times 0.95 + 24 \times 0.85 + 52 \times 0.75}{31 + 24 + 52} = 0.83$$

According to the above calculation, the results of each evaluation index of technical feasibility can be obtained, as shown in Table 8.

The remanufacturing technical feasibility of M1332B cylindrical grinder can be calculated by formula 5. The evaluation results are as follows.

$$
\begin{aligned}
T &= \alpha(\eta_d t_d + \eta_c t_c + \eta_i t_i + \eta_r t_r) \\
&= 1 \times (0.3937 \times 0.85 + 0.1374 \times 0.75 + 0.0752 \times 0.82 + 0.3937 \times 0.83) \\
&= 0.826
\end{aligned}
$$

According to the evaluation results, the remanufacturing technical feasibility of cylindrical grinder grinder is better, and the remanufacturing feasibility evaluation process of grinder can enter the next step.

The second step is to evaluate the remanufacturing economic feasibility of M1332B cylindrical grinder and calculate the remanufacturing cost to ensure that the remanufacturer is profitable. According to customer demand, the price of purchasing MKA1332 simple CNC cylindrical grinder with the same performance level needs RMB 320,000. The recycling value of the retired machine tool M1332B can be regarded as RMB 22,000. The expenditure of the purchased parts including the screw rod, CNC system and bearings is RMB 46,000. The cost of labor and consumables is RMB 37,800. The cost of factory management and redesign is RMB 9,000. The enterprise tax is RMB 9,200 and the enterprise profit is RMB 21,000. The remanufacturing economic feasibility of grinders can be calculated by formula 6 and the evaluation results are as follows.

$$f = \frac{C_1}{C_2} = \frac{22,000 + 46,000 + 37,800 + 9,000 + 9,200 + 21,000}{320,000} = 0.453$$

According to the value of $f$, the economic feasibility evaluation index $C$ can be obtained.

$$C = 1.8 - 2f = 1.8 - 2 \times 0.453 = 0.894$$

The economic feasibility evaluation results of remanufacturing show that the grinder has good economic feasibility, and the remanufacturer can obtain a larger profit, with remarkable economic benefits. The feasibility evaluation process of remanufacturing can enter the next step.

Finally, the feasibility of resources and environment for remanufacturing of M1332B cylindrical grinder is analyzed from the aspects of resource saving, energy saving and environmental protection. After cleaning and reprocessing, 96 parts of 107 waste parts can be used as

**Table 8. Technical feasibility evaluation results.**

| Index | Notions | Evaluation value |
|---|---|---|
| Ease of disassembly | $t_d$ | 0.85 |
| Feasibility of cleaning | $t_c$ | 0.75 |
| Feasibility of inspection and sorting | $t_i$ | 0.82 |
| Feasibility of part reconditioning | $t_r$ | 0.83 |
| Wear condition of guide rail | $\alpha$ | 1.00 |

remanufactured grinder parts for assembly, while 115 parts are needed for the same type of new grinder. The value of remanufactured grinder and the value of new grinder can be regarded as the same. According to formula 8, the resource utilization evaluation index of remanufactured grinder can be obtained.

$$R = \frac{N_u}{N_t} \times 1 = \frac{96}{115} = 0.834$$

The energy required in the manufacturing and remanufacturing process of grinding machine is electric energy. The electric energy required by the new grinding machine is about 675kwh, while the remanufacturing grinding machine only needs 56kwh. The pollutants discharged in the remanufacturing process of grinding machine are far lower than those discharged in the manufacturing process of new machine tools, so they fully meet the national environmental protection standards. According to formula 9, the evaluation index of remanufacturing energy saving rate can be obtained.

$$P = \frac{V_1 - V_2}{V_1} = \frac{675 - 56}{675} = 0.917$$

To sum up, the indicators of resource and environment feasibility are quantified, as shown in Table 9.

Subsequently, the remanufacturing resources environmental feasibility of M1332B cylindrical grinder can be calculated by formula 11, and the evaluation results are as follows.

$$
\begin{aligned}
E &= \eta(\omega_1 R + \omega_2 P) \\
&= 1 \times (0.75 \times 0.834 + 0.25 \times 0.917) \\
&= 0.855
\end{aligned}
$$

From the results of resource environment feasibility assessment, it can be seen that the remanufacturing of retired M1332B cylindrical grinder has better resource conservation and environmental friendliness. The grinder is suitable for remanufacturing and can maximize the reuse of waste resources.

## Process analysis of grinder remanufacturing

M1332B cylindrical grinder is mainly composed of bed, workbench, wheelhead frame, headframe, tailstock, hydraulic system, cooling system, protective cover, electrical system, etc. After disassembly, cleaning, testing and classification, the mechanical parts of the cylindrical grinder are repaired and reprocessed, the automatic operation effect has been achieved by adding the CNC system, servo motor and ball screw pair. The electrical system increases the variable-frequency drive to make the spindle speed of the machine tool reach the effect of stepless speed regulation. Adding magnetic coolant separator in cooling system can make ferromagnetic material separate automatically, keep cutting fluid clean, improve machining performance and reduce environmental pollution. After the retired parts with severe wear are replaced, the

**Table 9. Resources environment evaluation results.**

| Index | Notions | Evaluation value |
|---|---|---|
| Resource utilization rate | $R$ | 0.834 |
| Energy conservation rate | $P$ | 0.917 |
| Environmental protection rate | $\eta$ | 1.00 |

precision of grinder is restored to the factory standard, and the performance of machine tool can meet the production demand of customers.

After disassembly and cleaning, the loss of parts and precision of cylindrical grinder are analyzed, and the remanufacturing process plan is worked out. The remanufacturing process method and process of M1332B cylindrical grinder are as follows.

The repairing of grinder bed guide rail will cause different degree of wear and tear during the use of cylindrical grinder, which will affect the accuracy of machine tool processing. The guide rail of the bed casting can be repaired with casting adhesives when the wear is not serious. Then the guide rail grinder is used to grind the guide rail to the standard size. Finally, manual scraping of the bed guide rail is carried out. The purpose of scraping is to increase the actual contact area of the guide rail surface when working, and to store a small amount of lubricant in the scraping pit to reduce the friction of the moving contact surface and prolong the service life of the guide rail. The grinder integrated guideway is reconditioned as shown in Fig 8.

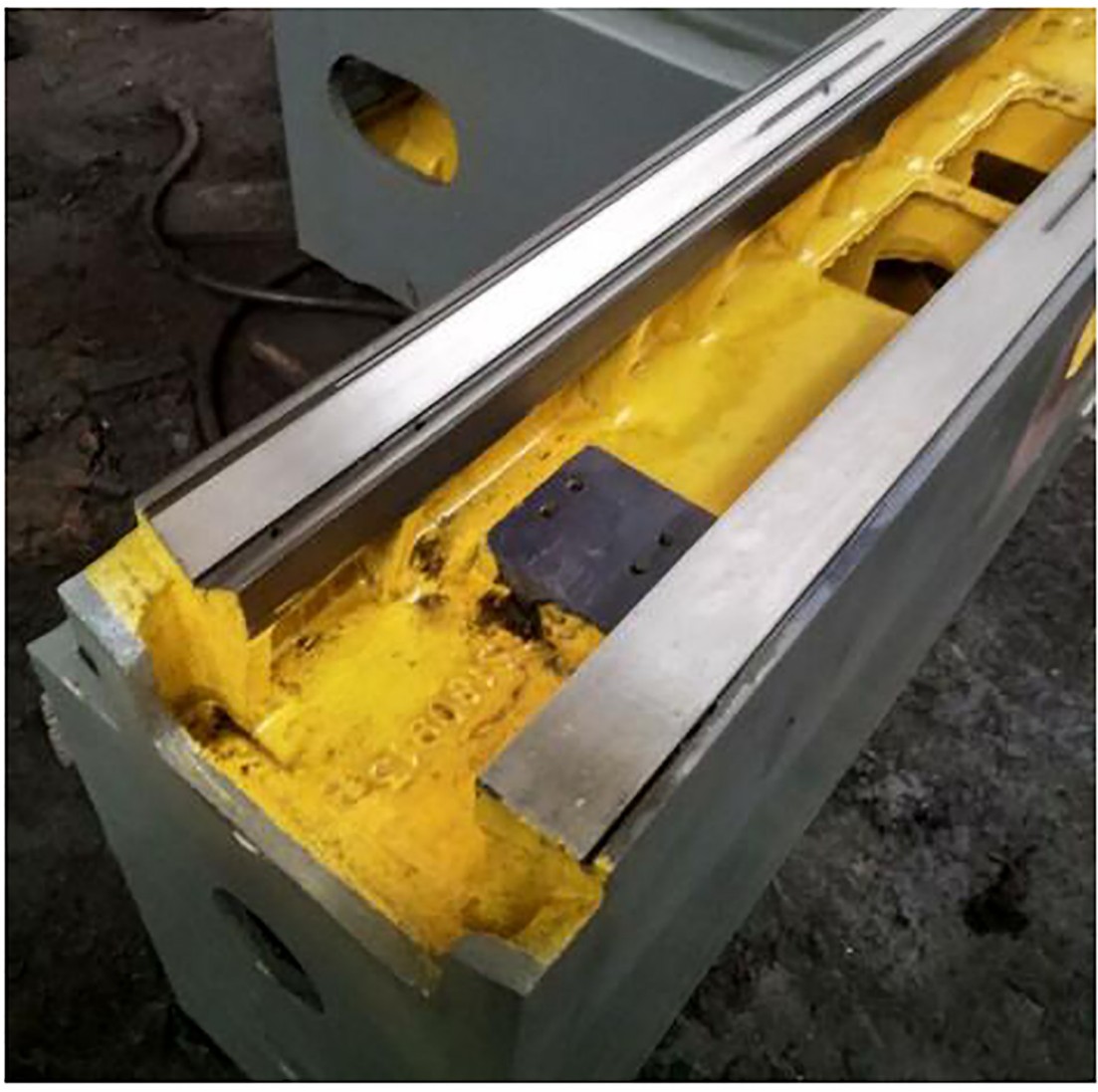

**Fig 8. Reconditioned grinding machine integrated guideway.**

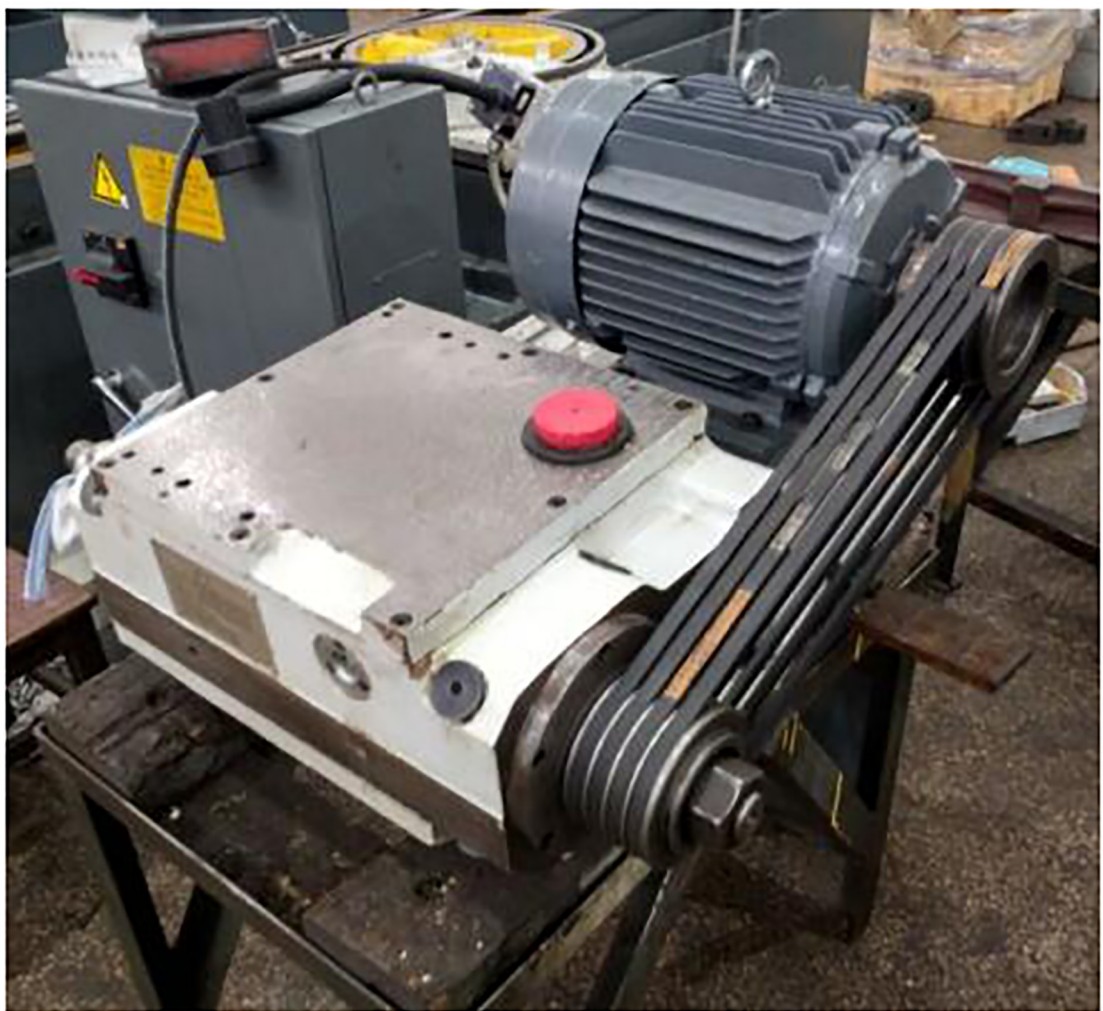

**Fig 9. Reconditioned grinding wheel frame spindle system.**

The grinding wheel frame spindle system is the core component of the cylindrical grinder. The remanufacturing results of the grinding wheel frame will directly affect the accuracy of the grinder. The remanufacturing of the grinding wheel frame includes the repair of the spindle and the replacement of the hydrodynamic and hydrostatic bearings. The surface defect of wheel frame spindle is repaired by coating method, and the roundness error of the spindle is less than 0.002 mm by grinding, the surface roughness can meet the requirements of Ra 0.2um. The gap between hydrodynamic bearing and spindle should meet the requirements of 0.015 mm to 0.025 mm. After reconditioning, the grinding wheel frame spindle system is shown in Fig 9.

Trapezoidal screw is used in the original drive mode of cylindrical grinder, which has been worn seriously. In order to ensure the motion accuracy and improve the motion flexibility, ball screw should be replaced. The improvement of grinder feed system is mainly to shorten the feed drive chain and drive the lead screw directly by servo motor, which can greatly reduce the error transmission between all levels of the drive chain. At the same time, servo motor

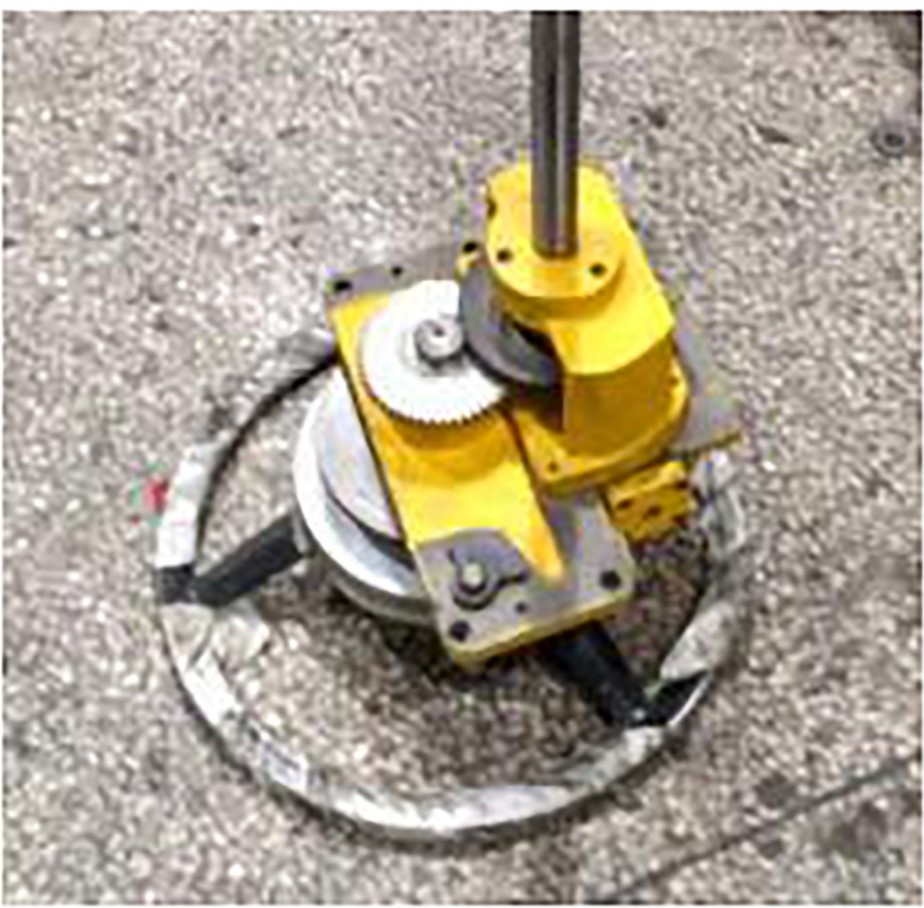

**Fig 10. Grinding machine feed system before reconditioning.**

drive can form semi-closed-loop control to improve machine tool processing accuracy. The feed system of cylindrical grinder before and after reconditioning are shown in Figs 10 and 11.

Retired M1332B cylindrical grinder is a manual grinder, which has no CNC system and has low control performance. It is difficult to meet the requirements of multi-batch and high efficiency processing parts. According to the principle of grinder remanufacturing combined with customer requirements, the electrical system remanufacturing scheme of cylindrical grinder is to install KT630G1 grinder CNC system, driver and control panel of Shanghai Capital NC Company, which can simultaneously realize rapid response and accurate control of output of each axis. Because the electrical control system of M1332B cylindrical grinder is completely different from that of CNC cylindrical grinder, and the electrical components of waste cylindrical grinder have been seriously aging, the original electrical control system of cylindrical grinder needs to be replaced, and the intensive power control box should be redesigned and manufactured. The intensive power control box and CNC system of M1332B cylindrical grinder before and after reconditioning are shown in Figs 12 and 13.

All parts of the remanufacturing grinder, including newly purchased parts and remanufactured parts, can be classified into eight categories according to their structure. The remanufacturing method and description of each part are shown in Table 10.

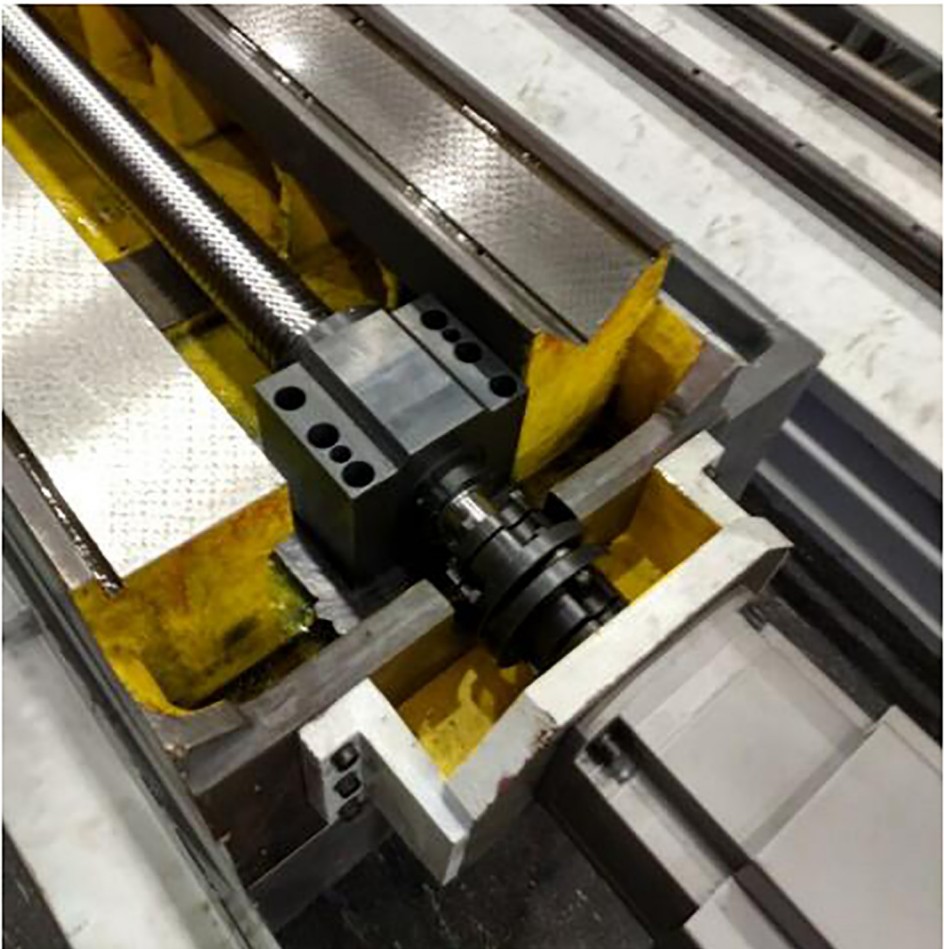

**Fig 11. Reconditioned grinding machine feed system.**

After repairing and reprocessing the parts, adjust the gap of the parts, scrape the joints of the parts, reassemble according to the assembly process, and restore the accuracy of the machine tool. Components assembled on remanufactured machine tools should meet the quality requirements. The remanufactured cylindrical grinder M1332B shall be inspected and checked according to the factory standard of the new machine tool. The results of comparison between the mechanical geometric accuracy of remanufactured cylindrical grinder and the factory standard are shown in Table 11.

The geometric accuracy of remanufactured cylindrical grinder meets the factory standard completely, and some indexes are even better. In addition, CNC system is added to remanufacturing grinder to realize automated operation. According to customer needs, intelligent and automation modules can be added in the follow-up. The state of remanufactured M1332B cylindrical grinder is shown in Fig 14.

Casting parts such as bed, workbench, grinding wheel frame, headframe and tailstock of waste M1332B cylindrical grinder and other parts with higher added value have been reused. The utilization rate of resource recycling is over 80% by weight, and the mechanical part of the machine tool has durability and stable performance. Especially for the bed castings, the longer

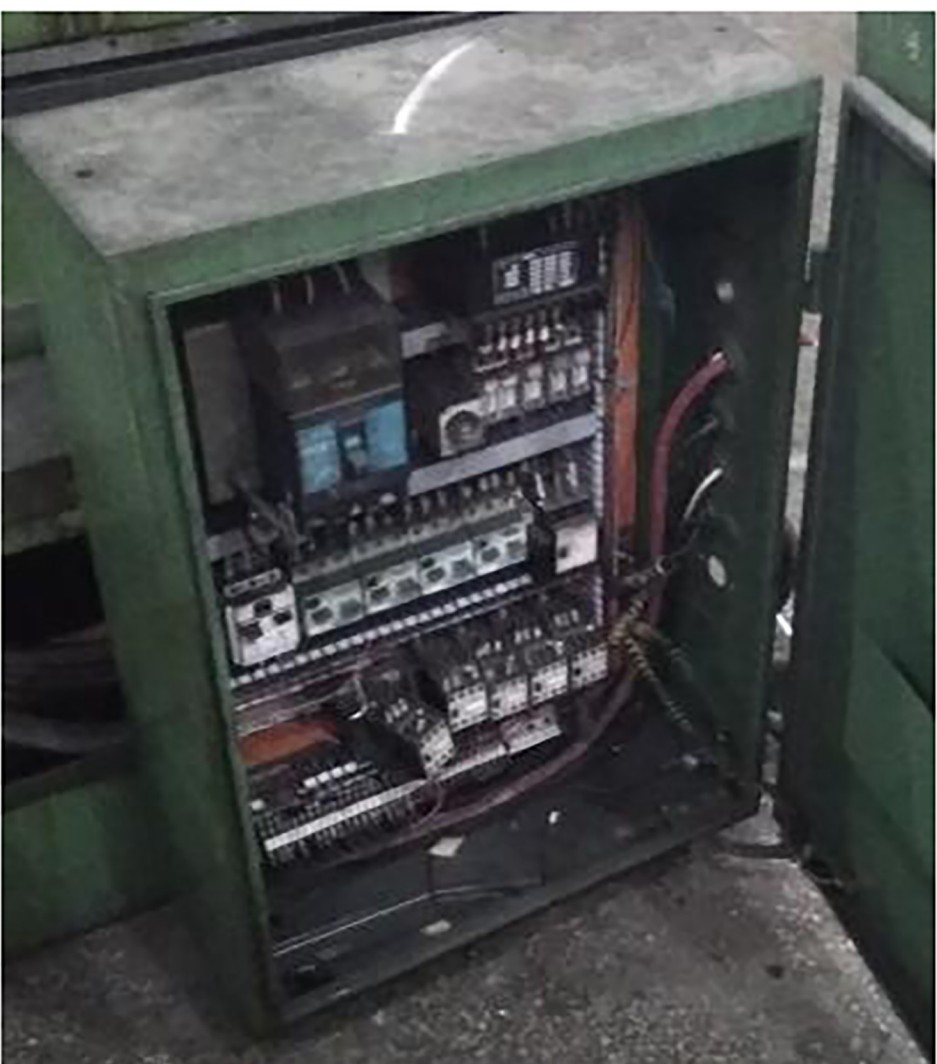

**Fig 12. Grinder intensive power control box before reconditioning.**

the aging time, the better the performance, the more stable and reliable the machine tool performance after remanufacturing.

Acceptance check of the working accuracy of the remanufactured cylindrical grinder M1332B. A 32x315mm alloy steel specimen was used. The rotational speed of the specimen was 80 to 130 r/min, and the workbench speed was 0.5 to 1.5m/min. As shown in Table 12, the working accuracy of remanufactured cylindrical grinder is compared with the factory standard.

According to the acceptance check of the working accuracy index of remanufactured M1332B cylindrical grinder, it is obvious that the remanufactured machine tool fully meets the factory standard of the new machine tool, and some of the indexes are better. The cost of remanufactured cylindrical grinder is over 50% less than that of newly purchased grinder, and the performance of machine tool is better than that of new machine tool.

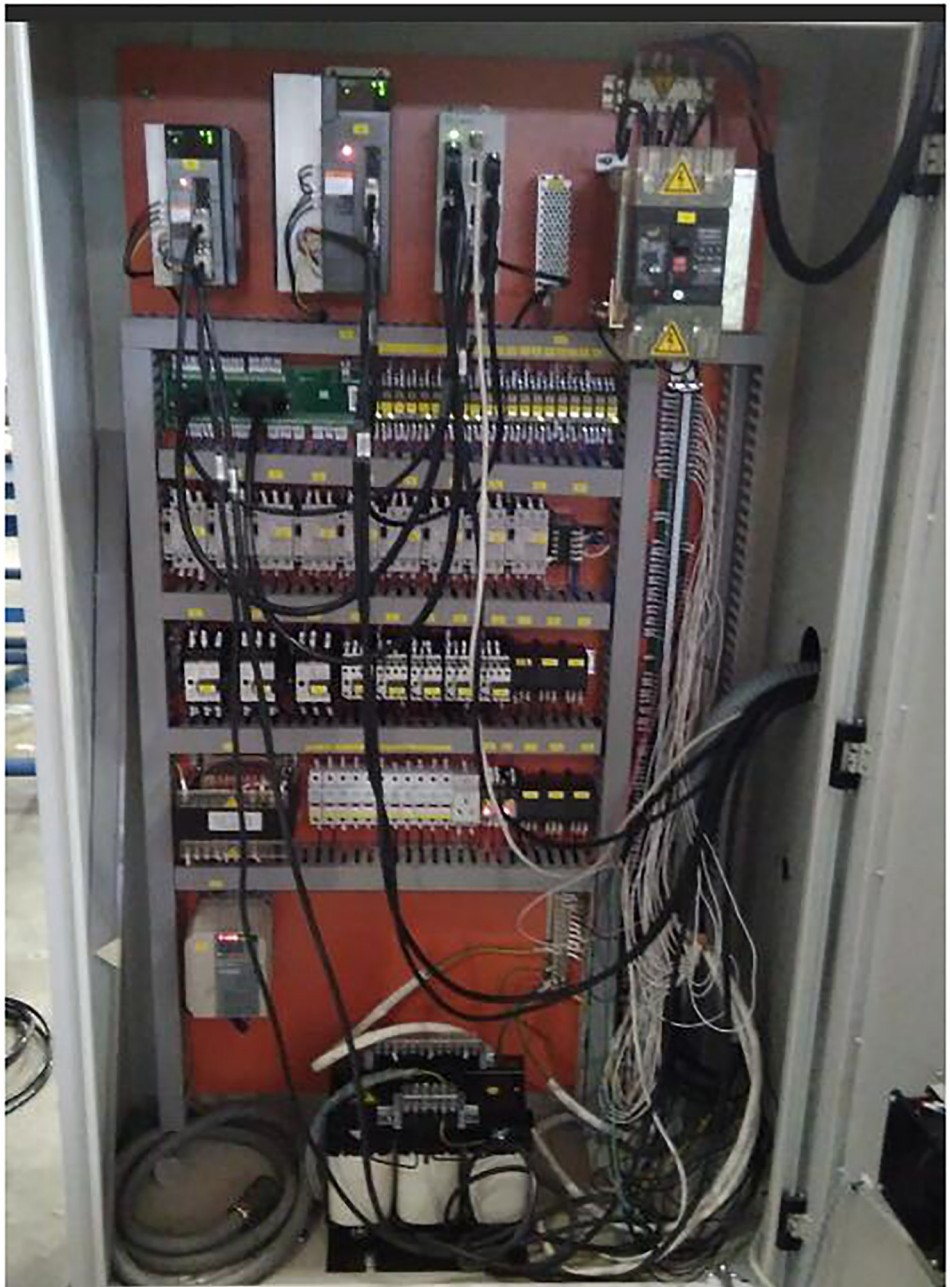

**Fig 13. Reconditioned grinder intensive power control box.**

## Comprehensive benefit analysis of grinder remanufacturing

The comprehensive benefit evaluation of remanufactured M1332B cylindrical grinder is carried out, the evaluation index factors are set up in Table 4. With using the scaling method of scale 1 to 9 in Table 2, the judgment matrix of the first-level index relative to the target level and the second-level index relative to the first-level index is determined. The matrices are shown in Tables 13–15.

**Table 10. Remanufacturing method for all parts of M1332B cylindrical grinder.**

| Number | Name of parts | Remanufacturing method | Description |
|---|---|---|---|
| 1 | bed | repairing | The bed can be reused directly and the guide rail surface needs to be grinded and manually scraped. |
| 2 | headframe | repairing | The bearings and seals in the headframe need to be replaced, other parts are reassembled after cleaning. |
| 3 | wheelhead frame | repairing | Spindle in the wheelhead frame need to be reprocessed and inspected and other parts are reassembled after cleaning. |
| 4 | tailstock | repairing | The remanufacturing process of tailstock is similar to that of headframe. |
| 5 | workbench | repairing | The guide rail surface of workbench needs to be grinded and manually scraped. |
| 6 | feed system | replacing | The CNC system including ball screw and servo motor replace the trapezoidal screw to improve grinding machine accuracy and efficiency. |
| 7 | cooling system | upgrading | Adding magnetic coolant separator in cooling system can keep cutting fluid clean and improve machining performance. |
| 8 | electrical system | upgrading | The electrical system increases the variable-frequency drive to make the spindle speed of the machine tool reach the effect of stepless speed regulation. |

The weight of individual secondary index and primary index for the comprehensive benefit evaluation of grinder remanufacturing is calculated, and the consistency is verified. For the first-level index judgment matrix $A$, the normalized results are as follows.

$W$ = (0.6817, 0.2364, 0.0819), Its maximum eigenvalue $\lambda_i$ = 3.0015, $CI$ = 0.0008, $RI$ = 0.58, $CR$ = 0.0014.

Similarly, the calculation and consistency test of the second-level weight indexes for the comprehensive benefit evaluation of grinder remanufacturing are as follows.

$W_1$ = (0.4073, 0.2112, 0.0748, 0.1948, 0.1119), Its maximum eigenvalue $\lambda_i$ = 5.0415, $CI$ = 0.0104, $RI$ = 1.12, $CR$ = 0.0093.

$W_2$ = (0.5714, 0.2857, 0.1429), Its maximum eigenvalue $\lambda_i$ = 3.0000, $CI$ = 0.0000, $RI$ = 0.58, $CR$ = 0.0000.

According to the experience, there are only two factors in environmental benefit index $u_3$, the weight of environmental benefit index can be obtained as $W_3$ = (0.7500, 0.2500).

Finally, according to the results of cylindrical grinder remanufacturing, twenty experts who have been engaged in machine tool remanufacturing field for a long time are invited to form an evaluation group to evaluate the remanufacturing cylindrical grinder comprehensively by voting. The specific evaluation results are shown in Table 16.

**Table 11. Acceptance results of remanufactured grinder geometric accuracy.**

| Number | Inspection items | Factory standard (mm) | Remanufactured grinder (mm) |
|---|---|---|---|
| G1 | Straightness of workbench moving (Z-axis) in Z-X horizontal plane | 0.015 | 0.012 |
| G2 | Straightness of grinding wheel frame moving (X-axis) in Z-X horizontal plane | 0.02 | 0.017 |
| G3 | Verticality of grinding wheel rack movement (X-axis) to table movement (Z-axis) | 0.015 | 0.013 |
| G4 | Parallelism of headframe spindle rotary axis to workbench movement (Z-axis) | 0.008 | 0.008 |
| G5 | Parallelism of taper hole axis of tailstock sleeve to workbench movement (Z-axis) | 0.008 | 0.006 |
| G6 | Radial runout of grinding wheel frame spindle | 0.005 | 0.005 |
| G7 | Parallelism of grinding wheel spindle axis to table movement (Z-axis) | 0.015 | 0.012 |
| G8 | Equal height of headframe spindle axis and grinding wheel spindle axis to reference plane (plane constituted by X-axis and Z-axis Moving) | 0.30 | 0.24 |

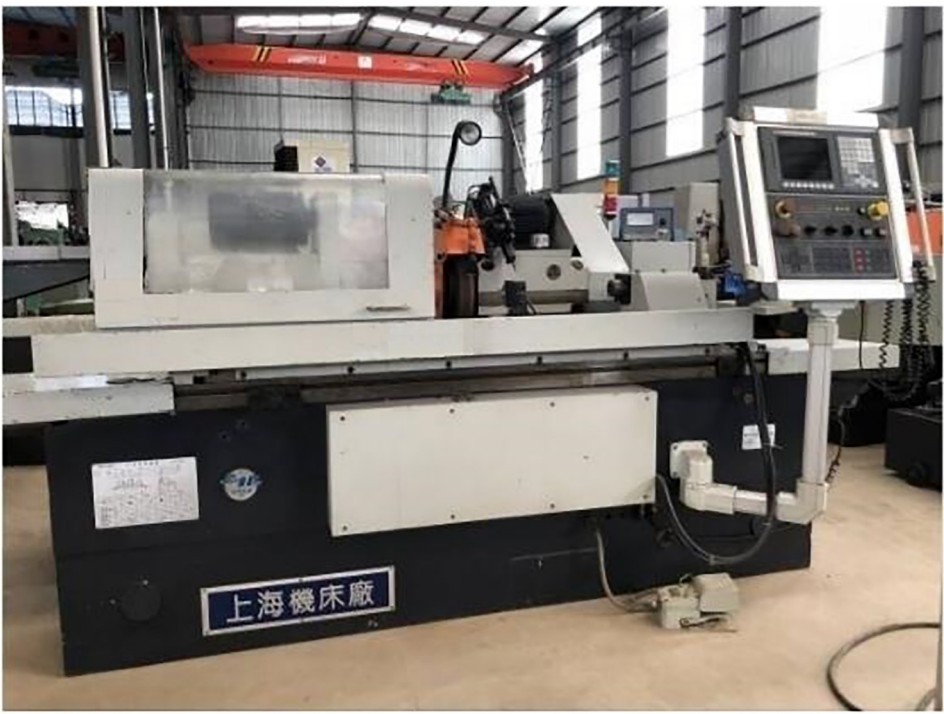

**Fig 14. The remanufactured M1332B cylindrical grinding machine.**

Table 12. Working accuracy of remanufacturing grinder.

| Number | Inspection items | Factory standard | Remanufacturing grinder |
|---|---|---|---|
| P1 | Roundness (um) | 1.5 | 1.1 |
| P2 | Uniformity of longitudinal Section diameter (um) | 3 | 2.3 |
| P3 | Surface roughness (Ra um) | 0.32 | 0.16 |

Table 13. First level index judgment matrix $A$.

| | Technical benefit ($u_1$) | Economic benefit ($u_2$) | Environmental benefit ($u_3$) |
|---|---|---|---|
| Technical benefit ($u_1$) | 1 | 3 | 8 |
| Economic benefit ($u_2$) | 1/3 | 1 | 3 |
| Environmental benefit ($u_3$) | 1/8 | 1/3 | 1 |

Table 14. Second level index judgment matrix $A_1$ of technique.

| | Modularization level ($u_{11}$) | Standardization level ($u_{12}$) | Ease of reassembly ($u_{13}$) | Machine upgrading ($u_{14}$) | Technical maturity ($u_{15}$) |
|---|---|---|---|---|---|
| Modularization level ($u_{11}$) | 1 | 2 | 5 | 2 | 4 |
| Standardization level ($u_{12}$) | 1/2 | 1 | 3 | 1 | 2 |
| Ease of reassembly ($u_{13}$) | 1/5 | 1/3 | 1 | 1/2 | 1/2 |
| Machine upgrading ($u_{14}$) | 1/2 | 1 | 2 | 1 | 1/2 |
| Technical maturity ($u_{15}$) | 1/4 | 1/2 | 2 | 1/2 | 1 |

**Table 15. Second level index judgment matrix $A_2$ of economics.**

| | Remanufacturing cost ($u_{21}$) | Remanufacturing profit ($u_{22}$) | Maintenance cost ($u_{23}$) |
|---|---|---|---|
| Remanufacturing cost ($u_{21}$) | 1 | 2 | 4 |
| Remanufacturing profit ($u_{22}$) | 1/2 | 1 | 2 |
| Maintenance cost ($u_{23}$) | 1/4 | 1/2 | 1 |

From Table 16, the membership degree of each influencing factor of the comprehensive benefit evaluation index of remanufacturing can be calculated, and the single factor membership degree matrix can be obtained. Taking technical benefit factors as an example, the technical membership matrix is obtained as follows.

$$R_1 = \begin{pmatrix} 0.45 & 0.40 & 0.10 & 0.05 \\ 0.25 & 0.55 & 0.15 & 0.05 \\ 0.30 & 0.45 & 0.15 & 0.10 \\ 0.60 & 0.30 & 0.10 & 0 \\ 0.70 & 0.20 & 0.10 & 0 \end{pmatrix}$$

According to formula 19.

$$B_1 = W_1 R_1 = \begin{pmatrix} 0.4073 \\ 0.2112 \\ 0.0748 \\ 0.1948 \\ 0.1119 \end{pmatrix}^{\mathrm{T}} \begin{pmatrix} 0.45 & 0.40 & 0.10 & 0.05 \\ 0.25 & 0.55 & 0.15 & 0.05 \\ 0.30 & 0.45 & 0.15 & 0.10 \\ 0.60 & 0.30 & 0.10 & 0 \\ 0.70 & 0.20 & 0.10 & 0 \end{pmatrix} = (0.4537, \; 0.3936, 0.1143, 0.0384)$$

Similar problem can be explained.

$$B_2 = W_2 R_2 = (0.5, 0.2929, 0.1571, 0.05)$$

$$B_3 = W_3 R_3 = (0.5\ 5, 0.3, 0.1375, 0.0125)$$

**Table 16. Expert evaluation results.**

| Index attribute | Evaluation results (number of votes) | | | |
|---|---|---|---|---|
| | Excellent evaluation | Good evaluation | General evaluation | Poor evaluation |
| Modularization level ($u_{11}$) | 9 | 8 | 2 | 1 |
| Standardization level ($u_{12}$) | 5 | 11 | 3 | 1 |
| Ease of reassembly ($u_{13}$) | 6 | 9 | 3 | 2 |
| Machine upgrading ($u_{14}$) | 12 | 6 | 2 | 0 |
| Technical maturity ($u_{15}$) | 14 | 4 | 2 | 0 |
| Remanufacturing cost ($u_{21}$) | 12 | 6 | 2 | 0 |
| Remanufacturing profit ($u_{22}$) | 7 | 6 | 5 | 2 |
| Maintenance cost ($u_{23}$) | 8 | 5 | 4 | 3 |
| Material reuse rate ($u_{31}$) | 12 | 5 | 3 | 0 |
| Energy conservation rate ($u_{32}$) | 8 | 9 | 2 | 1 |

According to formula 20 and the fuzzy comprehensive evaluation matrix $R = (B_1\ B_2\ B_3)$, the comprehensive benefit evaluation results of grinder remanufacturing are as follows.

$$B = WR = \begin{pmatrix} 0.6817 \\ 0.2363 \\ 0.0819 \end{pmatrix}^{\mathrm{T}} \begin{pmatrix} 0.4537 & 0.3936 & 0.1143 & 0.0384 \\ 0.5000 & 0.2929 & 0.1571 & 0.0500 \\ 0.5500 & 0.3000 & 0.1375 & 0.0125 \end{pmatrix} = (0.4725, 0.3621, 0.1263, 0.0390)$$

In order to transform the weight distribution of the four grades in the final evaluation set into a total score, the evaluation grades are quantified and assigned to the four grades by percentage system. Among them, the excellent value is 95 points, the good value is 85 points, the general value is 72 points and the poor value is 57 points. Then the calculation result of the comprehensive benefit evaluation value $Q$ for M1332B cylindrical grinder remanufacturing is as follows.

$$\begin{aligned} Q &= 0.4725v_1 + 0.3621v_2 + 0.1263v_3 + 0.039v_4 \\ &= 0.4725 \times 95 + 0.3621 \times 85 + 0.1263 \times 72 + 0.039 \times 57 \\ &= 86.9826 \end{aligned}$$

According to the comprehensive evaluation of remanufacturing benefit set $V$, it can be seen that the comprehensive benefit of remanufacturing M1332B cylindrical grinder is at a good level, close to the excellent level. The remanufacturing grinder has reached the expected level demanded by customers, and the waste resources and materials have been optimized.

## Conclusions

Grinding machine has high economic value and high processing precision, so the remanufacturing grinder has good economic, social and environmental benefits. However, due to the complex characteristics of grinder structure and the uncertainty of customer demand, there are many factors to be considered in the feasibility evaluation of grinder remanufacturing. This paper puts forward the feasibility evaluation model of grinder remanufacturing from three aspects of technical feasibility, economic feasibility and resource environmental feasibility, analyzes the quantitative methods of these evaluation standards, and proposes a relatively simple quantitative method to determine the evaluation value. Finally, the comprehensive benefit evaluation of grinding machine remanufacturing is used to determine the level of remanufacturing grinder. The case results show that the remanufacturing cost of the grinding machine is about 50% of the new machine tools of the same type, the utilization rate of used parts of the remanufacturing grinding machine reaches more than 80%, and the energy saving rate reaches 90%. After remanufacturing, the degree of automation and machining accuracy of cylindrical grinder have been significantly improved, and the remanufacturers can obtain considerable profits.

In the implementation of grinder remanufacturing, with the increase of information, the uncertain factors in the remanufacturing process also increase. The quantitative method of each evaluation index proposed in this paper does not consider the uncertainty of remanufacturing process, which is different from the actual situation. In future research, it is necessary to consider the uncertainty of the remanufacturing process and analyze the remanufacturing cases of different types of grinding machines so as to obtain more remanufacturing experience data and improve the performance and success rate of machine tool remanufacturing.

## Future research

In the future development process of machine tool remanufacturing, the remanufacturing process of grinders has developed from a small batch of manual manufacturing stage to a large batch of automated manufacturing stage. Intelligent technology will combine with remanufacturing technology to form intelligent remanufacturing to simplify remanufacturing process, including intelligent disassembly and cleaning technology, intelligent online monitoring technology and nondestructive testing technology. The application of advanced remanufacturing forming technology including 3D printing technology and nano material technology will be an important research direction in remanufacturing field and greatly improve the performance and service life of remanufacturing machine tools. Remanufacturing can not only prolong the life cycle of mechanical products, but also reflect the energy saving and emission reduction, which will become development trends for the future mechanical engineering.

## Author Contributions

**Conceptualization:** Tianbai Ling, Yongyi He.

**Data curation:** Tianbai Ling, Yongyi He.

**Funding acquisition:** Yongyi He.

**Methodology:** Yongyi He.

**Validation:** Tianbai Ling.

**Writing – original draft:** Tianbai Ling, Yongyi He.

**Writing – review & editing:** Tianbai Ling, Yongyi He.

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
