## [Decision Letter · Decision Letter 0]

2 Mar 2020

PONE-D-19-28523

The Remanufacturing Evaluation for Feasibility and Comprehensive Benefit of Retired Grinding Machine

PLOS ONE

Dear Dr Ling,

Thank you for submitting your manuscript to PLOS ONE. After careful consideration, we feel that it has merit but does not fully meet PLOS ONE’s publication criteria as it currently stands. Therefore, we invite you to submit a revised version of the manuscript that addresses the points raised during the review process.

We would appreciate receiving your revised manuscript by Apr 12 2020 11:59PM. To enhance the reproducibility of your results, we recommend that if applicable you deposit your laboratory protocols in protocols.io, where a protocol can be assigned its own identifier (DOI) such that it can be cited independently in the future. For instructions see: http://journals.plos.org/plosone/s/submission-guidelines#loc-laboratory-protocols

We look forward to receiving your revised manuscript.

Kind regards,

Yuvaraja Teekaraman, PhD

Academic Editor

PLOS ONE

Journal Requirements:

Please ensure that your manuscript meets PLOS ONE's style requirements, including those for file naming. The PLOS ONE style templates can be found at http://www.plosone.org/attachments/PLOSOne_formatting_sample_main_body.pdf and http://www.plosone.org/attachments/PLOSOne_formatting_sample_title_authors_affiliations.pdf

Reviewers' comments:

Reviewer's Responses to Questions

**Comments to the Author**

1. Is the manuscript technically sound, and do the data support the conclusions?

Reviewer #1: Partly

Reviewer #2: Partly

2. Has the statistical analysis been performed appropriately and rigorously? 

Reviewer #1: N/A

Reviewer #2: I Don't Know

3. Have the authors made all data underlying the findings in their manuscript fully available?

Reviewer #1: Yes

Reviewer #2: Yes

4. Is the manuscript presented in an intelligible fashion and written in standard English?

Reviewer #1: Yes

Reviewer #2: Yes

5. Review Comments to the Author

Reviewer #1: Dear editor in chief, Dear Authors,

This paper is good and interesting, written in suitable way, this paper described the concept of grinder remanufacturing. A comprehensive method for evaluating the feasibility of grinder remanufacturing is put forward from four aspects which is grinder structure, technical feasibility, economic feasibility and resource environment feasibility.

But I am not sure if the paper met the minimum requirement of plos one journal, as you know Q1 should has novelty or high contribution. Anyway, I have been written some comments that will enhanced the quality of paper to be more suitable.

1. Abstract needs some improvements by highlight the contributions and main findings.

2. The authors developed a strong methodological approach, but a theoretical background strengthening is recommended by adding more modern studies.

3. please standardize the writing of “fig or figure”.

4. The conclusion drawn from the study are acceptable but some of the conclusions are not mentioned. The conclusion should extract from result and discussion. Conclusions need improvements, could you highlight the main finding?

5. The contribution is not much clear, what is the difference with other researchers, could you provide comparison with others? Could you please highlight more the contribution of your work?

6. I highly recommend the authors also to focus their revised manuscript on the issues of: "Technology maturity" and "Economics" (or "Pricing") upon technologies discussed. This extra information could be extracted from either the existing 25 citations, or those that the authors will decide to include in their revised manuscript. This extra information should be gathered and given in quantitative or qualitative format, either in the "4. Discussion" or in the "5. Conclusion" sections.

7. Based on your experience, could you please add some future directions in this article in the conclusion part.

8. Please check the reference formatting writing according to Plos One journal standard.

Perhaps, the paper doesn’t present high level of new contribution or novelty, but I think the paper is could be accepted after making major corrections.

Reviewer #2: This is a good and interesting paper which aims to justify the remanufacturing and upgrades of heavy equipment which are in abundance in China. Result from this study can benefit other countries facing the same issue. The research output is timely and in-line with the concept of the Circular Economy which is accepted world-wide today.

The manuscript has the merit to be published however the following suggestions need to be considered by the authors prior to consideration for publication in PLOS ONE

1. Some parts of the manuscript refer to statistical data for example in paragraph 1, however the reference was not stated clearly.

2. The manuscript has to be proof read to correct some typographical and perhaps grammatical errors, for example:

In para 2, last line

Sect 1.2, first para

In Conclusion, should be AHP instead of APH

Page 20, line below formula 5

Sect 5, para 2, 1st line – Is it Three (name) or three (numbers)?

3. The manuscript addressed two strategies for retired grinding machines namely remanufacturing and upgrades. Upgrades contributed a major portion of the recovery, however upgrades was not addressed in the abstract, discussion and conclusion.

4. In Figure 5, how is T, C and E value of greater than or equal than 0.6 decided? Is it satisfactory or good?

5. In sect 5.2, it is not clear on what are the upgrades carried out on the electrical system and cooling system?

6. The remanufactured grinding machine comprises new parts and remanufactured parts. Some components are also upgraded. It would be clear or beneficial to the reader if the authors tabulated the parts of the grinding machine that are commonly remanufactured, replaced or upgraded.

7. For Table 6 on Technical feasibility evaluation, there is no detail justification on the value given for each technical aspect. For example, for ease of disassembly, the given evaluation value of 0.85 was not explained. What is the type of fasteners or whether the grinding machine is modular in order to justify the given value?

8. Similarly for Table 7 on Resources environment evaluation results. The evaluation value pertaining to the grinding machine used as the case example was not discussed.

6. PLOS authors have the option to publish the peer review history of their article (what does this mean?). If published, this will include your full peer review and any attached files.

Reviewer #1: No

Reviewer #2: No

---

## [Author Response · Author response to Decision Letter 0]

12 Apr 2020

Reviewer #1:

1.Abstract needs some improvements by highlight the contributions and main findings.

Response: I have improved the abstract and emphasized the contributions and main findings as follows.

Grinding is the last and most important process of parts processing, the purpose is to achieve high precision and surface roughness. Therefore, grinding machine has the characteristics of high added value, high technology content and great remanufacturing value. However, the evaluation of machine tool remanufacturing is based on imprecise and fuzzy information at present. The aim of this study is to present the remanufacturing evaluation for feasibility and comprehensive benefit of retired grinder. Firstly, according to the unique structure of grinder, the feasibility evaluation model of grinder remanufacturing is established, including technical feasibility criterion, economic feasibility criterion and resource environment feasibility criterion. Secondly, the comprehensive benefit evaluation model of remanufacturing grinder is established, in which the weight of each evaluation criterion is determined by Analytic Hierarchy Process (AHP). Finally, combined with the remanufacturing case of the cylindrical grinder, the evaluation method is verified and analyzed. The results show that the remanufacturing of the waste grinding machine through the feasibility evaluation can obtain better comprehensive benefits, and the remanufacturer can get considerable benefits and reduce the potential risks in the remanufacturing process.

2.The authors developed a strong methodological approach, but a theoretical background strengthening is recommended by adding more modern studies.

Response: In the introduction, literature review is added, and the theoretical background of machine tool remanufacturing research is introduced from line 131 to 167 of the manuscript. The details are as follows.

In the context of resource shortage and environmental pollution, remanufacturing as one of the well-known recovery methods for end-of-life products has become a research hotspot of various research institutions. The ability to design products for remanufacturing is usually owned by the original equipment manufacturers (OEM), who have control over both the design and remanufacturing phases of machine tool products. Not all machine tools are suitable for remanufacturing, only products with certain intrinsic value, long technology life cycle or high durability can be considered for remanufacturing. Detailed guidance for identifying remanufacturing candidates or components can be found in remanufacturing literature.

Modular design is built on the basis of functional analysis of a product with different functions or with the same function, but different performances and specifications. The use of modular design for remanufacturing is an effective way to perform the configuration of the product structure at the conceptual design stage. The modular division of machine tool can meet the requirements of remanufacturing of machine tool according to the characteristics of each stage of product life cycle, and the factors such as material, service life, maintainability, circular economy, efficiency and disassembly relationship are considered. In the process of modular design of machine tool products, module division is an important aspect of modular design, and it is of great significance for enterprises to implement machine tool remanufacturing.

a multi-objective optimization model with life cycle equilibrium and cost to maximize the recovery value of end-of-life (EOL) products was established. Case analysis showed that this scheme could effectively optimize the recovery value of EOL products and improve the economic benefits of remanufacturing. Different from the part remanufacturing such as components of photocopy machine and automotive parts, machine tool remanufacturing is the typical machine-based business, in which the used machine tools and parts are reused as the cores of remanufacturing to meet market demands by the processes of product innovation redesign, part reconditioning and machine upgrading. Not only the performance of the used machine tool can be restored to like-new condition, but also the energy efficiency, ecology efficiency and information function can be improved.

3.please standardize the writing of “fig or figure”.

Response: All writing standards are unified as Fig.

4.The conclusion drawn from the study are acceptable but some of the conclusions are not mentioned. The conclusion should extract from result and discussion. Conclusions need improvements, could you highlight the main finding?

Response: The conclusion is improved and the main findings are highlighted as follows.

Grinding machine has high economic value and high processing precision, so the remanufacturing grinder has good economic, social and environmental benefits. However, due to the complex characteristics of grinder structure and the uncertainty of customer demand, there are many factors to be considered in the feasibility evaluation of grinder remanufacturing. This paper puts forward the feasibility evaluation model of grinder remanufacturing from three aspects of technical feasibility, economic feasibility and resource environmental feasibility, analyzes the quantitative methods of these evaluation standards, and proposes a relatively simple quantitative method to determine the evaluation value. Finally, the comprehensive benefit evaluation of grinding machine remanufacturing is used to determine the level of remanufacturing grinder. The case results show that the remanufacturing cost of the grinding machine is about 50% of the new machine tools of the same type, the utilization rate of used parts of the remanufacturing grinding machine reaches more than 80%, and the energy saving rate reaches 90%. After remanufacturing, the degree of automation and machining accuracy of cylindrical grinder have been significantly improved, and the remanufacturers can obtain considerable profits.

5.The contribution is not much clear, what is the difference with other researchers, could you provide comparison with others? Could you please highlight more the contribution of your work?

Response: The differences from other researchers and the main contributions of the article are listed in the last paragraph of the literature review from line 160 to 167 of the manuscript, which is as follows.

From literature review and analysis, it is concluded that most of the relevant literature focuses on the modular design and redesign in new product development as well as the alternative selection of remanufacturing process and mainly include economic, technical, environmental and social factors. As a mechanical product with high value and complex structure, there is no suitable remanufacturing decision-making method, remanufacturing evaluation system and remanufacturing case analysis for selection in grinder remanufacturing. This paper establishes the feasibility evaluation and comprehensive benefit evaluation model of grinding machine remanufacturing from the combination of qualitative and quantitative methods to fill the gaps in the literature of current grinding machine remanufacturing decisions.

6.I highly recommend the authors also to focus their revised manuscript on the issues of: "Technology maturity" and "Economics" (or "Pricing") upon technologies discussed. This extra information could be extracted from either the existing 25 citations, or those that the authors will decide to include in their revised manuscript. This extra information should be gathered and given in quantitative or qualitative format, either in the "4. Discussion" or in the "5. Conclusion" sections.

Response: From line 450 to 482 of the manuscript, add the theory of innovative problem solving (TRIZ) to the contents of comprehensive benefit evaluation. From Tables 5 and 6 we can get that as for the prediction of the technical maturity of the remanufacturing grinder, it can be seen from CNIPA that there are 80315 patents related to machine tools in China from 2000 to 2019. From 2000 to 2006, the number of patents has been growing steadily. From 2007 to now, the number of patents has been growing rapidly. Machine tool products are developing towards numerical control, automation and intelligence.

In 2006, the number of CNC metal cutting machine tools in China reached 85,700. Since then, the production and CNC rate of machine tools have increased year by year. The CNC rate of machine tools has increased from 15.2% in 2006 to 40.1% in 2017.

According to the statistics of machine tool patents, machine tool output and CNC rate of machine tool, the number of machine tool patents increased steadily from 2000 to 2006, it can be seen that the manual grinder technology has entered the technical maturity stage in 2007 and gradually entered the withdrawal stage. Therefore, the return of the manual grinder to its original state can not obtain considerable benefits. The number of patents on intelligent and compound machine tools has been growing rapidly and the rate of CNC machine tools has been increasing since 2008,the CNC machine tool technology is still in the growth stage. Through the grinding machine remanufacturing, the CNC system is added to improve the automation of the machine tool and meet the needs of customers.

In Section feasibility analysis of grinder remanufacturing from line 580 to 586 of the manuscript, the pricing in the remanufacturing process is quantitatively analyzed as follows.

According to customer demand, the price of purchasing MKA1332 simple CNC cylindrical grinder with the same performance level needs RMB 320,000. The recycling value of the retired machine tool M1332B can be regarded as RMB 22,000. The expenditure of the purchased parts including the screw rod, CNC system and bearings is RMB 46,000. The cost of labor and consumables is RMB 37,800. The cost of factory management and redesign is RMB 9,000. The enterprise tax is RMB 9,200 and the enterprise profit is RMB 21,000. The remanufacturing economic feasibility of grinders can be calculated by formula 6 and the evaluation results are as follows.

According to the value of f, the economic feasibility evaluation index C can be obtained. The economic feasibility evaluation results of remanufacturing show that the grinder has good economic feasibility, and the remanufacturer can obtain a larger profit, with remarkable economic benefits. The feasibility evaluation process of remanufacturing can enter the next step.

7.Based on your experience, could you please add some future directions in this article in the conclusion part.

Response: The last paragraph of the conclusion from line 785 to 793 of the manuscript is future directions. 

In the future development process of machine tool remanufacturing, the remanufacturing process of grinders has developed from a small batch of manual manufacturing stage to a large batch of automated manufacturing stage. Intelligent technology will combine with remanufacturing technology to form intelligent remanufacturing to simplify remanufacturing process, including intelligent disassembly and cleaning technology, intelligent online monitoring technology and nondestructive testing technology. The application of advanced remanufacturing forming technology including 3D printing technology and nano material technology will greatly improve the performance and service life of remanufacturing machine tools. Remanufacturing can not only prolong the life cycle of mechanical products, but also reflect the energy saving and emission reduction, which will become an important direction for the future development of mechanical engineering.

8.Please check the reference formatting writing according to Plos One journal standard.

Response: The manuscript has been modified according to PLoS One journal standard. For details, please refer to the references of the manuscript.

Reviewer #2:

1.Some parts of the manuscript refer to statistical data for example in paragraph 1, however the reference was not stated clearly.

Response: These data are from literature including Prospects and Developing of Remanufacture Forming Technology and Experience and Enlightenment of foreign remanufacturing industry development, these two articles have been added to references [2] and [3].

2.The manuscript has to be proof read to correct some typographical and perhaps grammatical errors, for example: In para 2, last line, Sect 1.2, first para, In Conclusion, should be AHP instead of APH, Page 20, line below formula 5, Sect 5, para 2, 1st line – Is it Three (name) or three (numbers)?

 Response: All the APH have been changed to AHP. 

The grammatical errors in this sentence at the line below formula 5 from line 321 to 322 of the manuscript have been corrected as follows. When α=1, it means the bed guide rail can be recycled after repair; When α=0, it means the bed guide rail is severely worn or corroded and can not be recycled.

The three means three sets of new grinders, and all modified to three sets of new grinders from line 526 to 534 of the manuscript .

3.The manuscript addressed two strategies for retired grinding machines namely remanufacturing and upgrades. Upgrades contributed a major portion of the recovery, however upgrades was not addressed in the abstract, discussion and conclusion.

Response: In literature review from line 154 to 159 of the manuscript, Professor Xu Bing's definition of remanufacturing is added, which is as follows.

The important feature of remanufacturing is that the quality of remanufactured product is not inferior to the new product, and product performance has been significantly upgraded or improved. The cost of remanufactured products is only about 50% of new products. Compared with new products, remanufactured products can save about 60% of energy and 70% of materials. The environmental impact of remanufactured products is significantly reduced compared to the manufacture of new products. Restoring retired products to their original state is just repair or overhaul, not remanufacturing.

According to the statistics of table 5 and table 6 in comprehensive benefit evaluation from line 450 to 482 of the manuscript, the number of machine tool patents increased steadily from 2000 to 2006, it can be seen that the manual grinder technology has entered the technical maturity stage in 2007 and gradually entered the withdrawal stage. Therefore, the return of the manual grinder to its original state can not obtain considerable benefits. The number of patents on intelligent and compound machine tools has been growing rapidly and the rate of CNC machine tools has been increasing since 2008,the CNC machine tool technology is still in the growth stage. Through the grinding machine remanufacturing, the CNC system is added to improve the automation of the machine tool and meet the needs of customers.

In summary, upgrading is part of remanufacturing, and merely restoring the grinder to its original state is an overhaul. Through the grinding machine remanufacturing, the CNC system is added to improve the automation of the machine tool and meet the needs of customers.

4.In Figure 5, how is T, C and E value of greater than or equal than 0.6 decided? Is it satisfactory or good?

Response: The definition of the evaluation value is in lines 258 to 262 of the manuscript, the specific content is as follows.

The evaluation value including technical feasibility, economic feasibility, resource and environment feasibility is between 0 and 1. When the evaluation value is between 0.6 and 0.74, it means that the machine tool can be remanufactured. When the evaluation value is 0.75 to 0.89, it indicates good, and the remanufacturing of machine tool is profitable. When the evaluation value is more than 0.9, it means excellent, and machine tool remanufacturing can produce great benefits. 

5.In sect 5.2, it is not clear on what are the upgrades carried out on the electrical system and cooling system?

Response: The specific contents of upgrading the electrical system and cooling system is in Process analysis of grinder remanufacturingin from line 618 to 624 of the manuscript, which is as follows.

The automatic operation effect has been achieved by adding the CNC system, servo motor and ball screw pair. The electrical system increases the variable-frequency drive to make the spindle speed of the machine tool reach the effect of stepless speed regulation. Adding magnetic coolant separator in cooling system can make ferromagnetic material separate automatically, keep cutting fluid clean, improve machining performance and reduce environmental pollution. After the retired parts with severe wear are replaced, the precision of grinder is restored to the factory standard, and the performance of machine tool can meet the production demand of customers.

6.The remanufactured grinding machine comprises new parts and remanufactured parts. Some components are also upgraded. It would be clear or beneficial to the reader if the authors tabulated the parts of the grinding machine that are commonly remanufactured, replaced or upgraded.

Response: All parts of the remanufacturing grinder, including newly purchased parts and remanufactured parts, can be classified into eight categories according to their structure. The remanufacturing method and description of each part are shown in Table 10 of the manuscript.

7.For Table 6 on Technical feasibility evaluation, there is no detail justification on the value given for each technical aspect. For example, for ease of disassembly, the given evaluation value of 0.85 was not explained. What is the type of fasteners or whether the grinding machine is modular in order to justify the given value?

Response: The calculation method of each technical feasibility evaluation result in Table 8 is as follows from line 540 to 561 of the manuscript.

The first step in the remanufacturing process is to dismantle the used grinding machine M1332B. The main connecting parts of the machine are divided into three categories: screws, bolts and locating pins. The total number of screws is 326, with an average removal time of 5.7 seconds. The total number of bolts is 34, with an average removal time of 5.2 seconds. The total number of locating pins is 21, with an average removal time of 7.6 seconds. According to the expert's experience, the reference value of the dismantling time of the machine tool is 2,100 seconds, and the calculation results according to formula 1 are as follows.

Taking the above calculation results into formula 2, td = 0.85 can be obtained.

After disassembly, the grinder has a total of 107 parts, and 7 parts such as the body shell of the grinder need to be cleaned with chemical detergent spraying, the degree of difficulty is 0.7; The total number of spindles and other shaft parts is 8, which need to be baked, and the degree of difficulty is 0.4; The other 92 parts can be treated by blowing or brushing, and the degree of difficulty is 0.2. The value of tc can be obtained by formula 3. 

Most of the disassembled machine parts can be reused with only a small amount of wear, and the detection time is relatively small, so the evaluation result is defined as level B, which is ti=0.82.

A total of 107 machine parts need to be reprocessed after disassembly and cleaning. The total number of 31 casting parts is only a small amount of wear, and the repair success rate is 0.95; The number of 24 shaft and disk parts need to be precision reprocessed, with a success rate of 0.85; The other 52 parts include easily deformed parts such as protective cover, with a success rate of 0.75. The value of tr can be obtained by formula 4. 

8.Similarly for Table 7 on Resources environment evaluation results. The evaluation value pertaining to the grinding machine used as the case example was not discussed.

Response: The calculation method of resources environment evaluation results in Table 9 is as follows from line 592 to 605 of the manuscript.

After cleaning and reprocessing, 96 parts of 107 waste parts can be used as remanufactured grinder parts for assembly, while 115 parts are needed for the same type of new grinder. The value of remanufactured grinder and the value of new grinder can be regarded as the same. According to formula 8, the resource utilization evaluation index of remanufactured grinder can be obtained.

The energy required in the manufacturing and remanufacturing process of grinding machine is electric energy. The electric energy required by the new grinding machine is about 675kwh, while the remanufacturing grinding machine only needs 56kwh. The pollutants discharged in the remanufacturing process of grinding machine are far lower than those discharged in the manufacturing process of new machine tools, so they fully meet the national environmental protection standards. According to formula 9, the evaluation index of remanufacturing energy saving rate can be obtained.

---

## [Decision Letter · Decision Letter 1]

1 Jun 2020

The Remanufacturing Evaluation for Feasibility and Comprehensive Benefit of Retired Grinding Machine

PONE-D-19-28523R1

Dear Dr. Ling,

We are pleased to inform you that your manuscript has been judged scientifically suitable for publication and will be formally accepted for publication once it complies with all outstanding technical requirements.

With kind regards,

Yuvaraja Teekaraman, PhD

Academic Editor

PLOS ONE

Additional Editor Comments (optional):

Thank you for revising your paper for PLOS ONE. I am pleased to accept the paper for publication, since the criticisms of the reviewers have now been answered.

Reviewers' comments:

Reviewer's Responses to Questions

**Comments to the Author**

1. If the authors have adequately addressed your comments raised in a previous round of review and you feel that this manuscript is now acceptable for publication, you may indicate that here to bypass the “Comments to the Author” section, enter your conflict of interest statement in the “Confidential to Editor” section, and submit your "Accept" recommendation.

Reviewer #1: All comments have been addressed

Reviewer #2: All comments have been addressed

2. Is the manuscript technically sound, and do the data support the conclusions?

Reviewer #1: Yes

Reviewer #2: Yes

3. Has the statistical analysis been performed appropriately and rigorously? 

Reviewer #1: Yes

Reviewer #2: Yes

4. Have the authors made all data underlying the findings in their manuscript fully available?

Reviewer #1: Yes

Reviewer #2: Yes

5. Is the manuscript presented in an intelligible fashion and written in standard English?

Reviewer #1: Yes

Reviewer #2: No

6. Review Comments to the Author

Reviewer #1: The paper is interesting, All the comments have been addressed... just I would like to say that you have to insert the future study in the paper after the conclusion.

Reviewer #2: The revisions made are acceptable as it clarifies unambiguous statements in the earlier version.However, the paper must undergo proof reading and grammatical improvements in order to complement the improved technical content and overall quality of the paper.

The following are some examples of phrases that need improvement:

Line 140:

The modular division of machine tool can meet the requirements of remanufacturing of machine tool according to the characteristics of each stage of product life cycle, and the factors such as material, service life, maintainability, circular economy, efficiency and disassembly relationship are considered.

The sentence was not constructed in a good flow since the factors are lumped together; they are of different categories i.e material, design and dfX, economy model.

Line 148: is too long, require rephrasing and grammatical improvement:

Different from the part remanufacturing such as components of photocopy machine and automotive parts, machine tool remanufacturing is the typical machine-based business, in which the used machine tools and parts are reused as the cores of remanufacturing to meet market demands by the processes of product innovation redesign, part reconditioning and machine upgrading

Line 160: No references was made to support this claim:

From literature review and analysis, it is concluded that most of the relevant literature focuses on the modular design and redesign in new product development as well as the alternative selection of remanufacturing process and mainly include economic, technical, environmental and social factors

7. PLOS authors have the option to publish the peer review history of their article (what does this mean?). If published, this will include your full peer review and any attached files.

Reviewer #1: Yes: Mahir Faris Abdullah

Reviewer #2: No

---

## [Editor Report · Acceptance letter]

9 Jun 2020

PONE-D-19-28523R1 

The remanufacturing evaluation for feasibility and comprehensive benefit of retired grinding machine 

Dear Dr. Ling:

I'm pleased to inform you that your manuscript has been deemed suitable for publication in PLOS ONE. Congratulations! Your manuscript is now with our production department. 

Kind regards, 

on behalf of

Dr. Yuvaraja Teekaraman 

Academic Editor

PLOS ONE